# BALANCED NEURAL ODES: NONLINEAR MODEL ORDER REDUCTION AND KOOPMAN OPERATOR APPROXIMATIONS

**Julius Aka**
Chair of Mechatronics
University of Augsburg, Germany
`julius.aka@uni-a.de`

**Johannes Brunnemann & Jörg Eiden**
XRG Simulation GmbH
Hamburg, Germany
`{brunnemann,eiden}@xrg-simulation.de`

**Arne Speerforck**
Institute of Engineering Thermodynamics
Hamburg University of Technology, Germany
`speerforck@tuhh.de`

**Lars Mikelsons**
Chair of Mechatronics
University of Augsburg, Germany
`lars.mikelsons@uni-a.de`

## ABSTRACT

Variational Autoencoders (VAEs) are a powerful framework for learning latent representations of reduced dimensionality, while Neural ODEs excel in learning transient system dynamics. This work combines the strengths of both to generate fast surrogate models with adjustable complexity reacting on time-varying inputs signals. By leveraging the VAE's dimensionality reduction using a non-hierarchical prior, our method adaptively assigns stochastic noise, naturally complementing known NeuralODE training enhancements and enabling probabilistic time series modeling. We show that standard Latent ODEs struggle with dimensionality reduction in systems with time-varying inputs. Our approach mitigates this by continuously propagating variational parameters through time, establishing fixed information channels in latent space. This results in a flexible and robust method that can learn different system complexities, e.g. deep neural networks or linear matrices. Hereby, it enables efficient approximation of the Koopman operator without the need for predefining its dimensionality. As our method balances dimensionality reduction and reconstruction accuracy, we call it Balanced Neural ODE (B-NODE)[1]. We demonstrate the effectiveness of this methods on several academic and real-world test cases, e.g. a power plant or MuJoCo data.

## 1 INTRODUCTION

**Motivation:** Originally, this work is driven by challenges arising when using simulation models for optimization of complex technical systems. Increasing the fidelity of models generally results in a significant increase in computational expense. Consequently, simulation tasks such as real-time simulation for control purposes, long-term scenario analyses, or optimization processes requiring hundreds or thousands of simulations often become computationally prohibitive. For instance, in the automotive industry, detailed models of subsystems such as air conditioning, batteries, and engines must operate in real time for applications like predictive thermal control. Similarly, in the building sector, long-term simulations of detailed heat pump models are essential for design optimization and controller tuning. Therefore, attention to automated generation of faster surrogate and reduced-order models is growing. Reducing the model order directly lowers the computational expense of the solver to evaluate the underlying ODE. There exists special interest in linear surrogate models, as these allow the usage of a rich set of tools from control theory and efficient optimization algorithms.

This work describes a combination of (Dynamical) Variational Autoencoders (VAEs) with Neural ODEs, exploiting the VAE's information bottleneck for learning reduced-order dynamical systems.

---

[1]Our code is available under: `https://github.com/juliusaka/balanced-neural-odes`

**Neural ODEs for time-series modeling:** Neural ODEs (Chen et al., 2018) have gained significant attention in dynamic systems modeling. Extensions include neural controlled differential (Neural CDEs) equations (Kidger et al., 2020) and neural stochastic differential equations (Neural SDEs) (Liu et al., 2020). They excel in different areas, but first and foremost in time-series forecasting e.g. in geology (Shen et al., 2023), medicine (Hess et al., 2023; Shi et al., 2022), or as surrogate models in engineering (Legaard et al., 2022). To continuously integrate input signals, Neural CDEs (Kidger et al., 2020) have been used in several publications (Seedat et al., 2022; Hess et al., 2023; Wi et al., 2024). Controlled differential equations remain uncommon in dynamic system simulation, where state-space models are usually employed for the integration of inputs(Zamarreño & Vega, 1998), that can also be used with Neural ODEs (Rahman et al., 2022; Legaard et al., 2022). Neural ODEs can also be extended to include time-invariant modulators, i.e. parameterized dynamics (Lee & Parish, 2021; Auzina et al., 2023). Convergence of Neural ODE training can be improved by applying stochastic noise (Volokhova et al., 2019; Ghosh et al., 2020; Liu et al., 2020).

**Dynamical Variational Autoencoders** (DVAEs) are the extension of VAE (see 2.2) (Kingma & Welling, 2014) to sequential data (Girin et al., 2020). DVAEs usually include a separation of dynamic and fixed variables (Fraccaro et al., 2017; Sadok et al., 2024; Berman et al., 2024), and can further be distinguished by causal, anti- or noncausal inclusion of temporal dependencies (Girin et al., 2020). Deep Kalman Filters as a suitable subclass of causal DVAEs for system simulation employ a prediction-update logic for the dynamic latent variable and result in hierarchical priors (Krishnan et al., 2015; Fraccaro et al., 2017). To achieve measurable state reduction, our method requires a non-hierarchical prior. As a consequence, dynamics are part of the inference model, which additionally contains a temporal bottleneck, while the information bottleneck is represented by the latent space. To our knowledge, in DVAEs, probabilistic dynamic latent representations have never been modeled with Neural ODEs. On the other hand, LatentODEs (Rubanova et al., 2019; Shi et al., 2022; Dahale et al., 2023) combine VAE with Neural ODEs, but only assign the probability distribution to the initial latent state, and therefore fail in learning compact latent representations with input signals (4.1). For DVAEs, only the contribution of (Wi et al., 2024) proposed (independently from us) to continuously describe the variational parameters $\mu, \sigma$, but with a Neural CDE.

**Model Order Reduction (MOR) and Koopman Theory:** MOR deals with the challenge of generating models of reduced computational complexity from highly detailed simulation models (Schilders et al., 2008). This mostly manifests in a reduction of the system's state dimension. For linear systems, a variety of mathematical methods exists. Truncated Balanced Realization (Moore, 1981) is a prominent example and finds a state-space transformation that orders states by balancing its controllability and observability and truncates non-important. These techniques can also be used for nonlinear systems, but only in sufficiently small regions around a linearization point (Verriest, 2008). Koopman theory offers to overcome this by using observables $\boldsymbol{g}(\boldsymbol{x})$ of states that can be advanced in time with a linear operator, i.e. $\boldsymbol{g}(\boldsymbol{x}(t_{k+1})) = \mathcal{K} \cdot \boldsymbol{g}(\boldsymbol{x}(t_k))$, but with the drawback of this observable space being possibly infinite-dimensional (Koopman, 1931). Modern Koopman theory therefore deals with finding finite-dimensional approximations (Brunton et al., 2021). A popular approach for high-dimensional systems is Dynamic Mode Decomposition (DMD) (Schmid, 2010; Rowley et al., 2009). Several extensions exist, e.g. to inputs (DMDc, Proctor et al. (2016)), or to low-dimensional systems (Williams et al., 2015), where extended DMD lifts the dimensionality by e.g. radial basis functions (eDMDc, Korda & Mezić (2018)). Autoencoders (not VAEs) have also been applied to learn non-linear coordinate transformations to linear submanifolds (Lusch et al., 2018; Mardt et al., 2018). A different approach, modeling the dynamics with linear and non-linear components has been recently proposed by (Menier et al., 2023). A common obstacle to learning linear latent sub-manifolds is the choice of their dimensionality (Lusch et al., 2018; Menier et al., 2023; Mardt et al., 2018). We avoid this choice by weakly describing an information bottleneck with a balancing parameter $\beta$.

## 2 BACKGROUND

### 2.1 NEURAL ODES IN SYSTEM SIMULATION

In dynamic systems modeling, the state $\boldsymbol{x}$ is a vector of variables that is sufficient to reconstruct all other variables of a system at time $t$. Important for the remainder of this work, the choice of state variables is not unique (Schilders et al., 2008). The states are variables that appear as derivatives and must be determined by solving the ordinary differential equation (ODE) governing the dynamical

system (Cellier & Kofman, 2006). While this ODE is typically formulated using first principles, in the case of a Neural ODE, the right-hand side of the equation is modeled by a neural network $f_{\phi_{\text{NODE}}}$ (Chen et al., 2018),

$$\boldsymbol{x}'(t) = f_{\phi_{\text{NODE}}}\left(\boldsymbol{x}(t)\right), \boldsymbol{x}(t_0) = \boldsymbol{x}_0; \qquad \boldsymbol{x}(t = T) = \boldsymbol{x}_0 + \int_{t_0}^{t=T} f_{\phi_{\text{NODE}}}\left(\boldsymbol{x}(t)\right) dt.$$

The neural network $f_{\phi_{\text{NODE}}}(\boldsymbol{x}(t))$ describes the dynamics of the system. It maps to each state $\boldsymbol{x}$ a gradient $\boldsymbol{x}' = \frac{d\boldsymbol{x}}{dt}$ that describes the direction in which the state evolves. In other words, a vector field as in the example in Figure A.1 is learned.

For the analysis of dynamical systems, additional classes of variables are critically important: 1) time-varying external inputs $\boldsymbol{u}(t)$ excite the system and induce a response, e.g. a motor torque. 2) Time-invariant (physical) parameters $\boldsymbol{p}$ change the dynamic properties of the system, e.g. a mass. 3) Time-varying outputs $\boldsymbol{y}(t)$, e.g. a power, can be calculated as a function of the other variables $\boldsymbol{x}(t), \boldsymbol{p}, \boldsymbol{u}(t)$. By adding an output-function neural network $g_{\phi_{\text{Out}}}$ and the respective inputs, we get a state-space model (example in Figure A.1).

$$\boldsymbol{x}'(t) = f_{\phi_{\text{NODE}}}(\boldsymbol{x}(t), \boldsymbol{u}(t), \boldsymbol{p}), \qquad\qquad \boldsymbol{x}(t_0) = \boldsymbol{x}_0, \qquad (1)$$
$$\boldsymbol{y}(t) = g_{\phi_{\text{Out}}}(\boldsymbol{x}(t), \boldsymbol{u}(t), \boldsymbol{p}). \qquad\qquad (2)$$

We refer to this model (1-2) in the following as a *State Space Neural ODE (SS-NODE)* and train it by minimizing the MSE of predictions (23).

## 2.2 VARIATIONAL AUTOENCODERS FOR DIMENSIONAL REDUCTION

Variational Autoencoders (VAE) are frequently used to describe high-dimensional data $\boldsymbol{x}^{\mathcal{D}2}$ by using a smaller number of latent variables. VAEs assume a *prior distribution* on the latent variables and express the data probability as marginalization over the latent variable $\boldsymbol{z}$ (Kingma & Welling, 2019, 1.7)

$$p(\boldsymbol{x}^{\mathcal{D}}) = \int p(\boldsymbol{x}^{\mathcal{D}}|\boldsymbol{z})p(\boldsymbol{z})d\boldsymbol{z}.$$

For the prior, a multivariate Gaussian with zero mean and diagonal covariance one is typically assumed, i.e. $p(\boldsymbol{z}) = \mathcal{N}(\boldsymbol{z}; \boldsymbol{0}, \boldsymbol{I})$. To train the model $p(\boldsymbol{x}^{\mathcal{D}}|\boldsymbol{z})$, the intractability of the learning problem is resolved by approximating the true posterior probability $p(\boldsymbol{z}|\boldsymbol{x}^{\mathcal{D}})$ with an approximate model $q(\boldsymbol{z}|\boldsymbol{x}^{\mathcal{D}})$, called the *encoder* (Prince, 2023, 2.2). The model $p(\boldsymbol{x}^{\mathcal{D}}|\boldsymbol{z})$ is then the *decoder*. During training, the log-likelihood of the data $\log p(\boldsymbol{x}^{\mathcal{D}})$ shall be maximized. Due to the intractability, Kingma & Welling (2019, 2.2) derive a lower bound to the data likelihood, which is called the *evidence lower bound* (*ELBO*, derivation in A.3.1). We use the notation (Prince, 2023, 17.4.2)

$$\mathcal{G} = \mathbb{E}_{q(\boldsymbol{z}|\boldsymbol{x}^{\mathcal{D}})} \log \left[ \frac{p(\boldsymbol{x}^{\mathcal{D}}|\boldsymbol{z})}{q(\boldsymbol{z}|\boldsymbol{x}^{\mathcal{D}})} p(\boldsymbol{z}) \right] = \underbrace{\mathbb{E}_{q(\boldsymbol{z}|\boldsymbol{x}^{\mathcal{D}})} \log \left[ p(\boldsymbol{x}^{\mathcal{D}}|\boldsymbol{z}) \right]}_{\text{reconstruction quality}} - \beta \underbrace{D_{\text{KL}} \left[ q(\boldsymbol{z}|\boldsymbol{x}^{\mathcal{D}}) || p(\boldsymbol{z}) \right]}_{\text{information bottleneck}}, \quad (3)$$

with the term $\beta$ being added in $\beta$-VAE (Higgins et al., 2017). The functions $q(\boldsymbol{z}|\boldsymbol{x}^{\mathcal{D}}), p(\boldsymbol{x}^{\mathcal{D}}|\boldsymbol{z})$ are implemented with neural networks (A.2). Typically, the encoder network parameterizes a multivariate Gaussian distribution $f_{\phi_{\text{en}}}(\boldsymbol{x}^{\mathcal{D}}) = \boldsymbol{\mu}(\boldsymbol{z}), \boldsymbol{\sigma}(\boldsymbol{z})$, such that $q(\boldsymbol{z}|\boldsymbol{x}^{\mathcal{D}}) = \mathcal{N}(\boldsymbol{z}; \boldsymbol{\mu}(\boldsymbol{z}), \boldsymbol{\sigma}(\boldsymbol{z}))$. The decoder predicts the data $\hat{\boldsymbol{x}}^{\mathcal{D}} = f_{\phi_{\text{de}}}(\boldsymbol{z})$, with $\boldsymbol{z}$ following the conditional distribution of the encoder. For training, Kingma & Welling (2014) proposed to draw (one) Monte-Carlo sample of the posterior distribution by reparameterizing with an external noise variable, i.e. $\boldsymbol{z} = \boldsymbol{\mu}(\boldsymbol{z}) + \boldsymbol{\epsilon}\boldsymbol{\sigma}(\boldsymbol{z}); \boldsymbol{\epsilon} \sim \mathcal{N}(\boldsymbol{z}; \boldsymbol{0}, \mathbb{I})$, sustaining backpropagation ability.

Burgess et al. (2018) present $\beta$-VAE as a model with information bottleneck properties. Training a VAE maximizes the similarity between the data $\boldsymbol{x}^{\mathcal{D}}$ and the reconstruction $\hat{\boldsymbol{x}}^{\mathcal{D}}$, while the least amount of information is transmitted by the intermediate latent variable $\boldsymbol{z}$, which can be measured by the Kullback-Leibler Divergence $D_{\text{KL}}$. The balance toward higher reconstruction accuracy can be adjusted by lowering $\beta$. The data locality property of VAE describes that points close in data space are also encoded close in latent space (Burgess et al., 2018). Due to this, they excel in learning disentangled representations Higgins et al. (2017); Mathieu et al. (2019), and actually exhibit local orthogonality of the latent variables (Rolinek et al., 2019).

---

[2]$\boldsymbol{x}^{\mathcal{D}}$ is general data, $\boldsymbol{x}$ is the state.

To identify latent dimensions carrying a high information amount, KL divergence can be used. For the Gaussian prior with diagonal covariance, KL divergence can be analytically calculated (Odaibo, 2019). The overall $D_{\mathrm{KL}}$ is a sum of contributions from every latent channel

$$D_{\mathrm{KL}}\big(q_{\phi_{\mathrm{en}}}(\boldsymbol{z}|\boldsymbol{x}^{\mathcal{D}})||p(\boldsymbol{z})\big) = \sum_{i=0}^{n_{\boldsymbol{z}}} D_{\mathrm{KL},z_i} = \sum_{i=0}^{n_{\boldsymbol{z}}} -\frac{1}{2}\big[1 + \log(\sigma(z_i)^2) - \sigma(z_i)^2 - \mu(z_i)^2\big] \geq 0. \quad (4)$$

It forms a valley (Figure 1a) that ascends from $\mu(z_i) = 0, \sigma(z_i) = 1 \, (D_{\mathrm{KL},z_i} = 0)$. If a latent channel has $D_{\mathrm{KL},z_i} = 0$ for every input of the encoder, it transmits pure Gaussian noise, and therefore can not transmit any information. Therefore, only latent channels with $D_{\mathrm{KL},z_i} > 0$ can transmit information. Because of this, for dimensional reduction, a fixed prior is required. Hierarchical priors would learn characteristics of the data and therefore contradict using them as fixed standard of comparision.

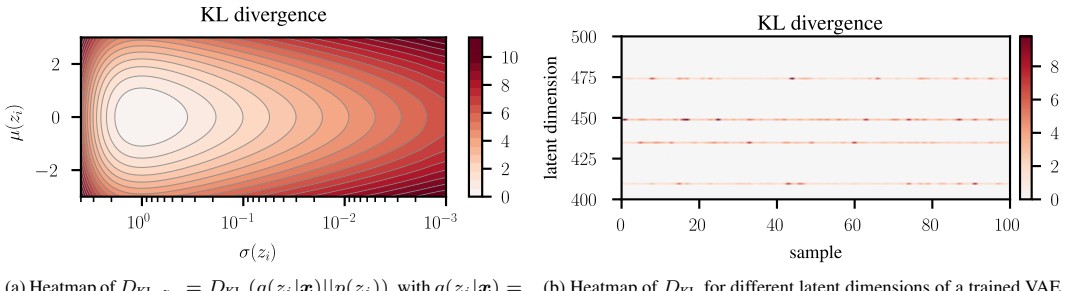

(a) Heatmap of $D_{\mathrm{KL},z_i} = D_{\mathrm{KL}}\left(q(z_i|\boldsymbol{x})||p(z_i)\right)$, with $q(z_i|\boldsymbol{x}) = \mathcal{N}(z;\mu(z_i),\sigma(z_i))$ and $p(z) = \mathcal{N}(\boldsymbol{z};0,1)$ (4).

(b) Heatmap of $D_{\mathrm{KL}}$ for different latent dimensions of a trained VAE for a given dataset: only 4 dimensions (in total 5) show values $\gg 0$.

Figure 1: Insights in the information transmission through the latent space. Figure 1b is the result of a $\beta$-VAE ($\beta = 1 \times 10^{-3}$) with a latent space of $n_{\boldsymbol{z}} = 512$ trained on a dataset of a thermal network model, generated by random sampling of 14 simulation model parameters (details in A.3.2).

To explore the latent space of a trained $\beta$-VAE, Figure 1b reveals that $D_{\mathrm{KL},z_i}$ is $\gg 0$ for only a small and fixed subset of the overall offered latent dimensionality. Ordering the dimensions for this example by their mean $D_{\mathrm{KL},z_i}$ reveals a significant drop of $D_{\mathrm{KL},z_i}$ from greater 1 to approximately 0 between the 5th and the 6th dimension (Figure A.3), indicating the first 5 dimensions contain the most important information, which was verified by experiment (Figure A.4a). This motivates to use the mean $D_{\mathrm{KL},z_i}$ per latent channel as a measure for active latent dimensions, and we choose for counting a value of $D_{\mathrm{KL},z_i} > 0.1$ (Figure A.4b).

## 3 METHOD: BALANCED NEURAL ODEs

Combining VAEs with State Space Models leverages VAEs' ability to generalize through an information bottleneck with a numerical pre-determined latent space that promotes data locality and latent orthogonality. The information bottleneck enables dimensional reduction without à priori defined latent dimensions, with the balance toward higher reconstruction accuracy adjusted by lowering the $\beta$-value. Additionally, the VAE's sampling-based robustness complements stability enhancements for Neural ODEs with stochastic noise. We call the proposed model *Balanced Neural ODE (B-NODE)*, as training results in a balance between state reduction and reconstruction quality.

In this work, we deal with continuous-time variables and their discrete-time samples. For that, a subscript $i$ denotes the value of a variable at time $t_i$, e.g. $\boldsymbol{x}_i = \boldsymbol{x}(t_i)$. We suppose (but are not limited to) equidistant time steps $\Delta t$ between observations $t_i$ and $t_{i+1}$. A sequence of observations in $[\boldsymbol{x}(t_i), \ldots, \boldsymbol{x}(t_j)]$, is denoted as $\boldsymbol{x}_{i:j}$. The initial time of a sequence is $t_0$, and the final time is $t_T$. For simplicity, inputs are considered constant in a time interval, $\boldsymbol{u}(t) = \boldsymbol{u}_i, t \in [t_i, t_{i+1}]$.

In the following, we derive the structure of a VAE that learns sequential representations of the data $\boldsymbol{x}^{\mathcal{D}} = [\boldsymbol{x}_{0:T}, \boldsymbol{y}_{0:T}, \boldsymbol{u}_{0:T}, \boldsymbol{p}]$ with latent variables $\boldsymbol{z} = [\boldsymbol{x}_{0:T}^z, \boldsymbol{u}_{0:T}^z, \boldsymbol{p}^z]$. We thereby follow the convention of Girin et al. (2020) to first present the model equations and then the implementation with neural networks.

## 3.1 INFERENCE MODEL AND GENERATIVE MODEL

In variational inference, the data $\boldsymbol{x}^{\mathcal{D}}$ is assumed to be generated from the "true" latent variables, thus it is necessary to infer these variables. We therefore describe the inference model as an encoding of all variables in data space,

$$q(\boldsymbol{z}|\boldsymbol{x}^{\mathcal{D}}) = q(\boldsymbol{p}^z|\boldsymbol{p})q(\boldsymbol{u}_{0:T}^z|\boldsymbol{u}_{0:T})q(\boldsymbol{x}_{0:T}^z|\boldsymbol{x}_{0:T}, \boldsymbol{y}_{0:T}, \boldsymbol{u}_{0:T}^z, \boldsymbol{p}^z), \tag{5}$$

where the time-invariant parameter vector (available from simulations) is simply encoded as $q(\boldsymbol{p}^z|\boldsymbol{p})$[3]. It is natural to assume that the control inputs are independent from $\boldsymbol{x}$ and $\boldsymbol{p}$, and thus can be encoded sequentially as

$$q(\boldsymbol{u}_{0:T}^z|\boldsymbol{u}_{0:T}) = \prod_{i=0}^{T} q(\boldsymbol{u}_i^z|\boldsymbol{u}_i). \tag{5a}$$

The inference model for the latent states is a combination of initial state observer and simulator:

$$q(\boldsymbol{x}_{0:T}^z|\boldsymbol{x}_{0:T}, \boldsymbol{y}_{0:T}, \boldsymbol{u}_{0:T}^z, \boldsymbol{p}^z) = \underbrace{q(\boldsymbol{x}_0^z|\boldsymbol{x}_{0:T}, \boldsymbol{y}_{0:T}, \boldsymbol{u}_{0:T}^z, \boldsymbol{p}^z)}_{\text{initial state observer}} \underbrace{\prod_{i=0}^{T-1} q(\boldsymbol{x}_{i+1}^z|\boldsymbol{x}_i^z, \boldsymbol{u}_i^z, \boldsymbol{p}^z)}_{\text{simulator}}. \tag{5b}$$

Having in mind that a surrogate model should simulate based on the inputs $\boldsymbol{x}_{\text{in, sim}}^{\mathcal{D}} = [\boldsymbol{x}_0, \boldsymbol{u}_{0:T}, \boldsymbol{p}]$, we prescribe a temporal information bottleneck starting from an initial latent state $\boldsymbol{x}_0^z$. With this, physical state properties are enforced: 1) All variables of the model at time $t_k$ can be described by a function of the set of latent states $\boldsymbol{x}_k^z$, latent inputs $\boldsymbol{u}_k^z$, and the latent parameters $\boldsymbol{p}^z$. 2) The state can be advanced in time with an integrator and inputs $\boldsymbol{u}^z(t)$ that arrive later in time.

For the initial state, we can argue that if we want to build a surrogate model of a physical simulation model, we have access to the state vector at time $t_0$. In dynamic systems the initial state is in general independent from the control input and the parameters, such that we have

$$q(\boldsymbol{x}_0^z|\boldsymbol{x}_{0:T}, \boldsymbol{y}_{0:T}, \boldsymbol{u}_{0:T}^z, \boldsymbol{p}^z) = q(\boldsymbol{x}_0^z|\boldsymbol{x}_0) \tag{5ba}$$

for the initial latent state, although integration of an initial state observer (identify $\boldsymbol{x}_0^z$ from $\boldsymbol{y}_{k_{\text{obs.}}:0}$) could also be accommodated.

Because $\boldsymbol{u}_{0:T}^z$ and $\boldsymbol{p}^z$ are inputs to the generative model and therefore known, the only data points of $\boldsymbol{x}^{\mathcal{D}}$ we are interested in approximating with a surrogate, are $\boldsymbol{y}^{\mathcal{D}} = [\boldsymbol{x}_{0:T}, \boldsymbol{y}_{0:T}]$. The generative model (decoder) is then simply a sequential decoding,

$$p(\boldsymbol{x}^{\mathcal{D}}|\boldsymbol{z}) \approx p(\boldsymbol{y}^{\mathcal{D}}|\boldsymbol{z}) = p(\boldsymbol{x}_{0:T}, \boldsymbol{y}_{0:T}|\boldsymbol{x}_{0:T}^z, \boldsymbol{u}_{0:T}^z, \boldsymbol{p}^z) = \prod_{i=0}^{T} p(\boldsymbol{x}_i, \boldsymbol{y}_i|\boldsymbol{x}_i^z, \boldsymbol{u}_i^z, \boldsymbol{p}^z). \tag{5}$$

Finally, the joint distribution of the observed and latent variables is

$$p(\boldsymbol{x}^{\mathcal{D}}, \boldsymbol{z}) = p(\boldsymbol{x}_{0:T}, \boldsymbol{y}_{0:T}|\boldsymbol{x}_{0:T}^z, \boldsymbol{u}_{0:T}^z, \boldsymbol{p}^z)p(\boldsymbol{p}^z)p(\boldsymbol{u}_{0:T}^z)p(\boldsymbol{x}_{0:T}^z). \tag{6}$$

To include state reduction features as discussed in 2.2, we set a (non-hierarchical ) Gaussian prior $p(\boldsymbol{z}) = p(\boldsymbol{p}^z), p(\boldsymbol{x}_{0:T}^z), p(\boldsymbol{u}_{0:T}^z) = \mathcal{N}(\boldsymbol{z}; \boldsymbol{0}, \boldsymbol{I})$. For using a trained B-NODE as a generative model, we need to control the data generation process. For that, we condition the distributions $p(\boldsymbol{p}^z), p(\boldsymbol{x}_{0:T}^z), p(\boldsymbol{u}_{0:T}^z)$ with the posteriors described in (5).

## 3.2 IMPLEMENTATION WITH DEEP NEURAL NETWORKS FOR MODEL ORDER REDUCTION

The parameter posterior density and control input posterior density (5a) are modeled as

$$q(\boldsymbol{p}^z|\boldsymbol{p}) = \mathcal{N}(\boldsymbol{p}^z; \boldsymbol{\mu}(\boldsymbol{p}^z), \boldsymbol{\sigma}(\boldsymbol{p}^z)), \qquad (\boldsymbol{\mu}(\boldsymbol{p}^z), \boldsymbol{\sigma}(\boldsymbol{p}^z)) = \boldsymbol{f}_{\phi_{\text{en,p}}}(\boldsymbol{p}) \tag{7}$$

$$q(\boldsymbol{u}_i^z|\boldsymbol{u}_i) = \mathcal{N}(\boldsymbol{u}^z; \boldsymbol{\mu}(\boldsymbol{u}_i^z), \boldsymbol{\sigma}(\boldsymbol{u}_i^z)), \qquad (\boldsymbol{\mu}(\boldsymbol{u}_i^z), \boldsymbol{\sigma}(\boldsymbol{u}_i^z)) = \boldsymbol{f}_{\phi_{\text{en,u}}}(\boldsymbol{u}_i). \tag{8}$$

The latent initial state posterior density (5ba) for $\boldsymbol{x}_0^z$ is likewise implemented. The inference of the full latent state posterior density sequence $\boldsymbol{x}_{1:T}^z$ (5b) is implemented with a Neural ODE

$$q(\boldsymbol{x}_0^z|\boldsymbol{x}_0) = \mathcal{N}(\boldsymbol{x}_0^z; \boldsymbol{\mu}(\boldsymbol{x}_0^z), \boldsymbol{\sigma}(\boldsymbol{x}_0^z)), \qquad (\boldsymbol{\mu}(\boldsymbol{x}_0^z), \boldsymbol{\sigma}(\boldsymbol{x}_0^z)) = \boldsymbol{f}_{\phi_{\text{en,}x_0}}(\boldsymbol{x}_0) \tag{9}$$

$$q(\boldsymbol{x}_{i+1}^z|\boldsymbol{x}_i^z, \boldsymbol{u}_i^z, \boldsymbol{p}^z) = \mathcal{N}(\boldsymbol{x}_{i+1}^z; \boldsymbol{\mu}(\boldsymbol{x}_{i+1}^z), \boldsymbol{\sigma}(\boldsymbol{x}_{i+1}^z))$$

$$\begin{bmatrix} \boldsymbol{\mu}(\boldsymbol{x}_{i+1}^z) \\ \boldsymbol{\sigma}(\boldsymbol{x}_{i+1}^z) \end{bmatrix} = \begin{bmatrix} \boldsymbol{\mu}(\boldsymbol{x}_i^z) \\ \boldsymbol{\sigma}(\boldsymbol{x}_i^z) \end{bmatrix} + \int_{t_i}^{t_{i+1}} \boldsymbol{f}_{\phi_{\text{NODE}}}(\boldsymbol{\mu}(\boldsymbol{x}^z(t)), \boldsymbol{\sigma}(\boldsymbol{x}^z(t)), \boldsymbol{u}^z(t), \boldsymbol{p}^z)dt, \tag{10}$$

---

[3]$\boldsymbol{p}, \boldsymbol{p}^z$ refer to physical parameters, $p(\bullet)$ to a probability density function, $\phi_{\square}$ to neural network parameters.

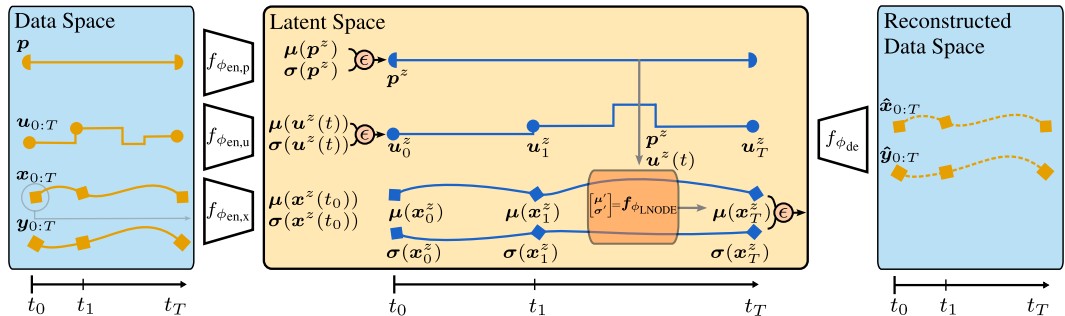

Figure 2: Scheme of the Balanced Neural ODE.

that propagates the distribution parameters $\boldsymbol{\mu}(\boldsymbol{x}^z(t)), \boldsymbol{\sigma}(\boldsymbol{x}^z(t))$ through time. Figure 2 shows the scheme of the resulting model.

For $\boldsymbol{f}_{\phi_{\text{LNODE}}}(\bullet)$, different configurations are possible. We can distinguish *constant variance* and *dynamic variance* state reduction

$$\boldsymbol{f}_{\phi_{\text{LNODE}}}(\bullet) = \begin{bmatrix} \boldsymbol{\mu}(\boldsymbol{x}_k^z) \\ \boldsymbol{\sigma}(\boldsymbol{x}_k^z) \end{bmatrix}' = \begin{bmatrix} \boldsymbol{f}_{\phi_{\text{LNODE,const}}}(\boldsymbol{\mu}(\boldsymbol{x}^z(t)), \boldsymbol{u}^z(t), \boldsymbol{p}^z) \\ 0 \end{bmatrix} \tag{11}$$

$$\boldsymbol{f}_{\phi_{\text{LNODE}}}(\bullet) = \begin{bmatrix} \boldsymbol{\mu}(\boldsymbol{x}_k^z) \\ \boldsymbol{\sigma}(\boldsymbol{x}_k^z) \end{bmatrix}' = \begin{bmatrix} \boldsymbol{f}_{\phi_{\text{LNODE,dyn},\mu}}(\boldsymbol{\mu}(\boldsymbol{x}^z(t)), \boldsymbol{u}^z(t), \boldsymbol{p}^z) \\ \boldsymbol{f}_{\phi_{\text{LNODE,dyn},\sigma}}(\boldsymbol{\mu}(\boldsymbol{x}^z(t)), \boldsymbol{\sigma}(\boldsymbol{x}^z(t)), \boldsymbol{u}^z(t), \boldsymbol{p}^z) \end{bmatrix}. \tag{12}$$

For the latter, the mean is independent of the variance. Like this, information with state properties can not be encoded in the variance part of the state vector.

Stability and convergence of Neural ODEs can be improved by "smoothing" the ODE's vector field with application of stochasticity during training (Volokhova et al., 2019; Ghosh et al., 2020; Liu et al., 2020). The VAE framework includes adaptively learned variance variables, that are applied as stochastic noise during training (Kingma & Welling, 2019). To bring both together, we propose to set for the unsampled inputs of $\boldsymbol{f}_{\phi_{\text{LNODE}}}(\boldsymbol{\mu}(\boldsymbol{x}^z(t)), \boldsymbol{\sigma}(\boldsymbol{x}^z(t)))$ during training

$$\boldsymbol{\mu}^*(\boldsymbol{x}^z(t)) = \boldsymbol{\mu}(\boldsymbol{x}^z(t)) + \epsilon\boldsymbol{\sigma}(\boldsymbol{x}^z(t)), \qquad \epsilon \sim \mathcal{N}(\boldsymbol{0}, \boldsymbol{I}) \tag{13}$$

$$\boldsymbol{\sigma}^*(\boldsymbol{x}^z(t)) = \boldsymbol{\sigma}(\boldsymbol{x}^z(t)) + \alpha_\sigma \cdot \epsilon\boldsymbol{\sigma}(\boldsymbol{x}^z(t)), \qquad \epsilon \sim \mathcal{N}(\boldsymbol{0}, \boldsymbol{I}). \tag{14}$$

To learn the state reduction in $\boldsymbol{f}_{\phi_{\text{LNODE}}}$, sampling of the mean state $\boldsymbol{\mu}(\boldsymbol{x}^z(t))$ is required. If a state $x_i^z$ is inactive, the model learns $\boldsymbol{\mu}(x_i^z) = 0$ and $\boldsymbol{\sigma}(x_i^z) = 1$ for that state. Then the reparameterized state (13) is distributed unconditioned, $\mu^*(x_i^z(t)) = \mathcal{N}(0, 1)$, i.e. no information transmission is possible and the model learns to ignore that dimension. For $\alpha_\sigma$, a small value should be chosen.

The generative model (decoder) $p(\boldsymbol{x}_i, \boldsymbol{y}_i | \boldsymbol{x}_i^z, \boldsymbol{u}_i^z, \boldsymbol{p}^z)$ (5) is represented with (A.4.3)

$$\hat{\boldsymbol{x}}_i, \hat{\boldsymbol{y}}_i = \boldsymbol{f}_{\phi_{\text{de}}}(\boldsymbol{x}_i^z, \boldsymbol{u}_i^z, \boldsymbol{p}^z), \tag{15}$$

and evaluated after sampling from the conditional distributions of the function inputs. For accurate generative usage of B-NODEs, we set all noise variables $\boldsymbol{\epsilon} = \boldsymbol{0}$.

### 3.3 ADAPTATIONS TO LEARN KOOPMAN OPERATOR APPROXIMATIONS

Learning Koopman operator approximations is as simple as setting for $\boldsymbol{f}_{\phi_{\text{LNODE}}}$ a linear model with parameters $\phi = \{A_{i,i}, B_{i,i}\}$ instead of neural networks. For constant-variance state reduction (11)

$$\begin{bmatrix} \boldsymbol{\mu}(\boldsymbol{x}_k^z) \\ \boldsymbol{\sigma}(\boldsymbol{x}_k^z) \end{bmatrix}' = \begin{bmatrix} \boldsymbol{A}_{\mu,\mu} & \boldsymbol{0} \\ \boldsymbol{0} & \boldsymbol{0} \end{bmatrix} \cdot \begin{bmatrix} \boldsymbol{\mu}(\boldsymbol{x}_k^z) \\ \boldsymbol{\sigma}(\boldsymbol{x}_k^z) \end{bmatrix} + \begin{bmatrix} \boldsymbol{B}_{\mu,u} \\ \boldsymbol{0} \end{bmatrix} \cdot \boldsymbol{u}^z(t). \tag{16}$$

and for dynamic-variance state reduction (12)

$$\begin{bmatrix} \boldsymbol{\mu}(\boldsymbol{x}_k^z) \\ \boldsymbol{\sigma}(\boldsymbol{x}_k^z) \end{bmatrix}' = \begin{bmatrix} \boldsymbol{A}_{\mu,\mu} & \boldsymbol{0} \\ \boldsymbol{A}_{\sigma,\mu} & \boldsymbol{A}_{\sigma,\sigma} \end{bmatrix} \cdot \begin{bmatrix} \boldsymbol{\mu}(\boldsymbol{x}_k^z) \\ \boldsymbol{\sigma}(\boldsymbol{x}_k^z) \end{bmatrix} + \begin{bmatrix} \boldsymbol{B}_{\mu,u} \\ \boldsymbol{B}_{\sigma,u} \end{bmatrix} \cdot \boldsymbol{u}^z(t). \tag{17}$$

Opposed to Autoencoder approaches for learning Koopman operators (Brunton et al., 2021), the latent dimension must only be implicitly set with the $\beta$-value (see 2.2).

### 3.4 TRAINING

Following the approach of Girin et al. (2020) to explicitly represent the sampling dependencies, the data likelihood and KL divergence terms of the ELBO (3) can be expanded to (derived in A.4.1)

$$\mathbb{E}_{q(\boldsymbol{z}|\boldsymbol{x}^{\mathcal{D}})} \log \left[ p(\boldsymbol{x}^{\mathcal{D}}|\boldsymbol{z}) \right] = \mathbb{E}_{q(\boldsymbol{u}_{0:T}^{z}, \boldsymbol{p}^{z}|\boldsymbol{u}_{0:T}, \boldsymbol{p})} \left[ \mathbb{E}_{q(\boldsymbol{x}_{0:T}^{z}|\boldsymbol{x}_0, \boldsymbol{u}_{0:T}^{z}, \boldsymbol{p}^{z})} \sum_{i=0}^{T} \log p(\boldsymbol{x}_i, \boldsymbol{y}_i|\boldsymbol{x}_i^{z}, \boldsymbol{u}_i^{z}, \boldsymbol{p}^{z}) \right]$$

$$D_{\mathrm{KL}} \left[ q(\boldsymbol{z}|\boldsymbol{x}^{\mathcal{D}})||p(\boldsymbol{z}) \right] = D_{\mathrm{KL}} \left[ q(\boldsymbol{p}^{z}|\boldsymbol{p})||p(\boldsymbol{p}^{z}) \right] + D_{\mathrm{KL}} \left[ q(\boldsymbol{u}_{0:T}^{z}|\boldsymbol{u}_{0:T})||p(\boldsymbol{u}_{0:T}^{z}) \right]$$
$$+ \mathbb{E}_{q(\boldsymbol{u}_{0:T}^{z}, \boldsymbol{p}^{z}|\boldsymbol{u}_{0:T}, \boldsymbol{p})} \left( D_{\mathrm{KL}} \left[ q(\boldsymbol{x}_{0:T}^{z}|\boldsymbol{x}_0, \boldsymbol{u}_{0:T}^{z}, \boldsymbol{p}^{z})||p(\boldsymbol{x}_{0:T}^{z}) \right] \right). \quad (18)$$

As we simultaneously learn latent embedding and dynamics in latent space, the first step in training a B-NODE is finding initially stable ODE systems. Proven strategies for that are growing horizon (from small to long prediction horizons) and multiple shooting (splitting sequences in multiple, smaller ones) approaches. Then, we observe robust training behavior for constant variance B-NODEs with moderate $\beta$ values (i.e. $\beta = 0.1, 0.01$), which is from our experience more stable than State Space Neural ODEs with the same data.

## 4 APPLICATIONS: B-NODES FOR STATE REDUCTION

### 4.1 ACADEMIC USE CASE: DISCRETIZED HEAT FLOW

First, we illustrate state reduction, generalization, and performance of B-NODEs with a simple 1D heat flow model. It consists of 16 cells describing the temperature field of a stick whose ends are excited with external input signals (A.5.2). To generate datasets from physical models, we employ a sampling strategy that aims to capture all relevant state-input combinations (A.5.1).

Figure 3a shows the original temperature field generated by the heat flow model $\boldsymbol{x}_{0:T}$, its state-

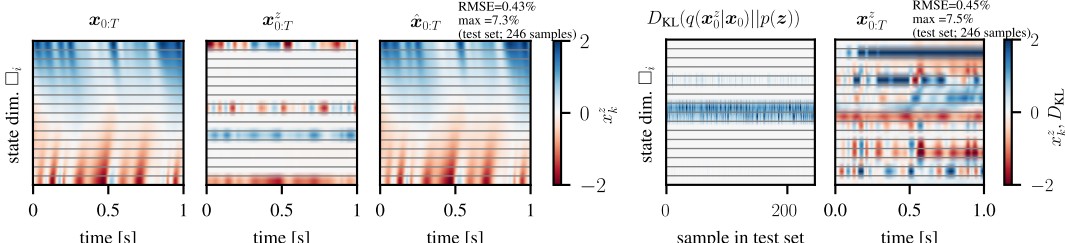

(a) Reconstruction of temperature field from latent states and latent inputs for one sample in test set, B-NODE with constant variance.

(b) Even though $D_{\mathrm{KL}}=0$ for several dimensions, Latent ODE with inputs $\boldsymbol{u}^{z}(t)$ does not achieve state reduction.

Figure 3: Comparison of state reduction of B-NODE and vanilla Latent ODE (Rubanova et al., 2019) for discretized heat flow example.

reduced representation in latent space $\boldsymbol{x}_{0:T}^{z}$ and the reconstruction of the temperature field with the decoder $\hat{\boldsymbol{x}}_{0:T}$ with a RMSE of $0.4\,\%$ (loss-values normalized by respective means). Taking the MOR perspective, we learn a nonlinear embedding of the full state space on a latent sub-manifold with reduced dimensionality, requiring only 4 instead of 16 states, sufficient to globally describe the full state space. This embedding is described bi-directional with an encoder and a decoder. The dimensionality results from the balance between state reduction and reconstruction quality, and is modified with $\beta$ (Figure 4a). As in Figure 3a, the state reduction is constant over the whole time sequence. This also holds when transforming the weakly applied information bottleneck (by KL divergence) to a hard one (by masking latent channels), see Figure A.10. Because of that, latent space dimensionalities are not important hyperparameters, given they are large enough.

Naively adding inputs $\boldsymbol{u}^{z}(t)$ to Latent ODE (Rubanova et al., 2019) as in Figure 3b leads to failing state reduction. The Latent ODE propagates a reparameterized initial latent state $\boldsymbol{x}_0^{z}$ and does not allow to enforce state reduction at times $t > 0$.

Instead of learning the system's response, we learn the system dynamics $(\boldsymbol{x}^{z})'$ as a vector field depending on $\boldsymbol{x}^{z}, \boldsymbol{u}^{z}, \boldsymbol{p}^{z}$. This enables generalization to input signals unseen in training, for example, step inputs (Figure 4b).

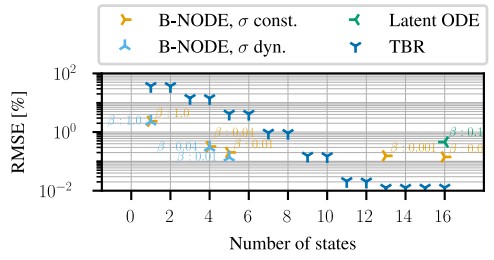
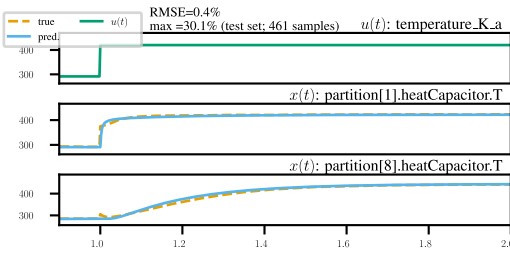

(a) Comparison between prediction quality of B-NODEs and Truncated Balanced Realization (TBR).

(b) Generalization: model prediction for sampled step inputs; sample with highest RMSE.

Figure 4: Generalization and comparison of methods for discretized heat flow example.

As the discretized heat flow model is described by a linear system, we can apply the model order reduction technique Truncated Balanced Realization (TBR), which finds a balance of well-observable and controllable states (Schilders et al., 2008). Our method consistently outperforms TBR (Figure 4a), for example, for 4 states the B-NODE has an RMSE of $0.3\,\%$ vs. $14.2\,\%$ for TBR.

## 4.2 REAL-WORLD USE CASE: SURROGATE MODEL OF A THERMAL POWER PLANT

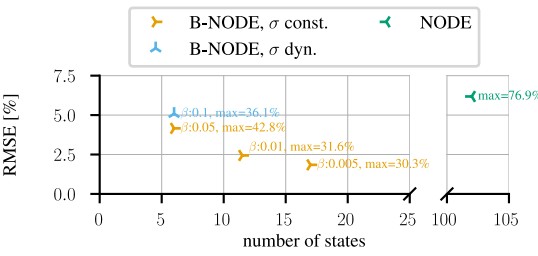

Figure 5: Error comparison key quantities of interest (outputs) for surrogates of power plant.

To test the surrogate qualities with a complex model, we employ a dataset generated by a power plant simulation model (Vojacek et al., 2023), of which we predict 102 states and 6 output signals, reacting on one input signal. Figure 5 compares RMSE and maximum error of different B-NODEs and a SS-NODE as surrogates. The B-NODE with $\beta = 0.05$ and constant variance exhibits the smallest reconstruction error of all models while only requiring 17 states instead of 102.

Taking a look into the latent space of the B-NODE with dynamic variance, Figure 6a exhibits a reduction of variance at times with high dynamical behavior. A possible explanation could be that dynamics are encoded by states, while steady-state behavior is modeled by the decoder. However, we can perform uncertainty quantification in data space by sampling and generating multiple trajectories.

Finally, we make a performance comparison (Figure 6b). Most significantly, the B-NODE facilitates a $32\times$ speed-up in computation time on CPU and a $103\times$ speed-up on GPU for batch simulation. It also simulates several factors faster than SS-NODE. Both models were evaluated using an adaptive steps size ODE solver. As a consequence of model order reduction and training with stochastic noise (13-14) B-NODE requires 10 times less ODE function evaluations for integration than SS-NODE.

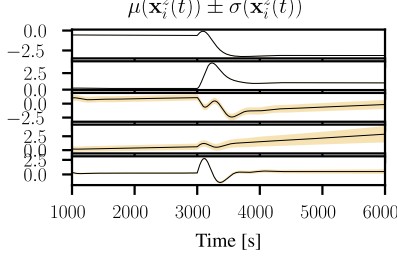

(a) Latent state trajectories for B-NODE with dynamic variance ($\beta = 0.1$), ordered by mean $D_{KL}$, tested with step input.

| Scenario | # Sim. | Time | ODE eval. |
|---|---|---|---|
| *Baseline* with *FMPy* (FMPy, 2023) | | | |
| CPU, 1 core | 1 | 16 s | |
| CPU, 16 core | 1024 | 720 s | |
| *Surrogate Methods* with *torchdiffeq* (Chen, 2018) | | | |
| B-NODE (CPU) | 1 | 0.5 s, 32× | 1012 |
| SS-NODE (CPU) | 1 | 3.7 s, 4× | 10 033 |
| B-NODE (GPU) | 1024 | 7.0 s, 103× | 4316 |
| SS-NODE (GPU) | 1024 | 68.0 s, 10× | 14 654 |

(b) Computation time comparison. Technical details in A.5.4.

Figure 6: Application results for the power plant simulation model.

### 4.3 BENCHMARK

Table 1 provides a benchmark of the proposed methodology for different datasets (details for Mu-JoCo and Waterhammer in A.5.7-A.5.8). As shown, B-NODE only marginally increases the prediction error compared to other methods but vastly reduces the required state dimension for all datasets. Not controlling the latent state embedding as for B-NODE with $\beta = 0.0$ and Latent Neural ODE leads to higher errors or failing prediction if inputs are incorporated.

Table 1: Benchmark results (RMSE and state dimension) for nonlinear surrogates. (A.5.9).

|  | SHF | | Power Plant | | MuJoCo | | Water Hammer | |
|---|---|---|---|---|---|---|---|---|
|  | RMSE | dim. | RMSE | dim. | RMSE | dim. | RMSE | dim. |
| SS-NODE | 0.01 | 16 | 0.06 | 102 | 0.43 | 27 | 0.02 | 400 |
| B-NODE $\beta = 0.0$ | 0.01 | 16 | 18.2 | 128 | 0.51 | 128 | 0.05 | 512 |
| B-NODE $\beta = 0.01$ | 0.02 | 5 | 0.02 | 11 | 0.28 | 25 | 0.03 | 8 |
| B-NODE $\beta = 0.1$ | 0.04 | 4 | 0.04 | 6 | 0.50 | 11 | 0.05 | 4 |
| Latent NODE $\beta = 0.1$ | 0.05 | 16 | 0.14 | 128 | 0.43 | 128 | 0.06 | 512 |

## 5 APPLICATIONS: B-NODES FOR KOOPMAN OPERATOR APPROXIMATION

### 5.1 ACADEMIC USE CASE

An example for a nonlinear system that can be linearized by adding a state is (Brunton et al., 2021)

$$\begin{bmatrix} x_1 \\ x_2 \end{bmatrix}' = \begin{bmatrix} ax_1 \\ b \cdot (x_2 - x_1^2) \end{bmatrix}, \qquad \begin{bmatrix} x_1 \\ x_2 \\ x_{3,add.} \end{bmatrix}' = \begin{bmatrix} a & 0 & 0 \\ 0 & b & -b \\ 0 & 0 & 2a \end{bmatrix} \cdot \begin{bmatrix} x_1 \\ x_2 \\ x_{3,add.} \end{bmatrix}. \qquad (19)$$

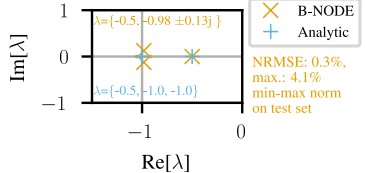

Figure 7: Eigenvalues of analytic solution and B-NODE.

System dynamics of linear systems are determined by the eigenvalues (here $\lambda = \{a, b, 2a\}$). If two systems have the same eigenvalues, they represent the same dynamics. The B-NODE can choose different states than the analytic solution, but projects them back into the state space of the analytic model with a linear decoder. We trained a constant variance B-NODE as in 3.3 with linear layers with a dataset generated by sampling initial values with parameters $a = $-0.5, $b$=-1 for different values of $\beta$ and offered a latent dimensionality of dim($\boldsymbol{x}^z$)=64.

For $\beta$ of 0.01, we found 3 active dimensions ($D_{\mathrm{KL}}(x_i^z) > 0.1$), just as the analytic solution has. The approximated and analytic eigenvalues lie at almost the same position (Figure 7) and therefore represent the same dynamics.

### 5.2 REAL-WORLD USE CASE: LINEARIZED MODEL OF A THERMAL POWER PLANT

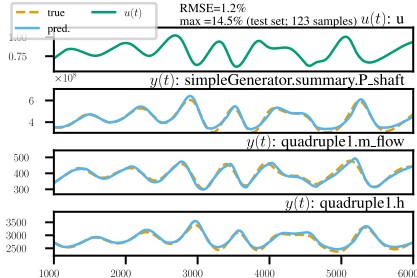

Figure 8: Reconstruction sample of B-NODE ($\beta = 0.01$) with highest error and 25 linear states (original 102 nonlinear; offered dim($\boldsymbol{x}^z$)=1024).

A model with a full linear path from model input to model output is desirable for model-predictive control (MPC) applications. It can be learned by setting linear models with an additive bias for the control encoder $\boldsymbol{f}_{\phi_{\mathrm{en,u}}}$ and the decoder $\boldsymbol{f}_{\phi_{\mathrm{de}}}$. Then, only the state encoder has non-linearities and facilitates a nonlinear state transformation into a linear latent space that can approximately represent the dynamics. We trained constant variance B-NODEs with $\beta \in \{0.1, 0.01, 0.001\}$ for this use case with dim($\boldsymbol{x}^z$)=dim($\boldsymbol{u}^z$)=1024 to predict all 102 state variables and 6 output variables of interest for the thermal power plant simulation model.

The best model achieved a RMSE of $1.2\,\%$ and a maximum error of $14.5\,\%$ on the selected outputs (normalized

by mean) (Figure 8), while requiring 25 states. Interestingly, even though we only have one control input signal $u(t)$, the latent control dimension increases to 18 inputs. We observed that for short prediction horizons, the approximation quality is better, which fits the short-horizon model requirement of MPC.

## 5.3 BENCHMARK

A benchmark with other popular linear surrogate methods is given in Table 2. For the Power Plant, B-NODE achieves the same accuracy than eDMDc, completely avoiding observable function tuning. The Small Grid model A.5.6 as an example for MOR of computational fluid dynamics (CFD) can be learned with higher accuracy than with DMDc.

Table 2: Benchmark results (RMSE and state dimension) of linear surrogates (A.5.10).

|  | Power Plant | | Small Grid | |
| --- | --- | --- | --- | --- |
|  | RMSE | dim. | RMSE | dim. |
| B-NODE $\beta = 0.01$ | 0.20 | 25 | 0.48 | 30 |
| B-NODE $\beta = 0.1$ | 0.20 | 12 | 0.52 | 17 |
| eDMDc | 0.20 | 45 | - | - |
| DMDc | - | - | 0.60 | 2 |

## 6 DISCUSSION

Table 3: Comparison of B-NODE and other surrogate methods.

|  | Control inputs | Nonlinear Model-order reduction | Koopman operator approximation | Low-resolution data | Uncertainty Quantification |
| --- | --- | --- | --- | --- | --- |
| B-NODE | ✓ | ✓ | ✓ | ✓ | ✓ |
| SS-NODE | ✓ | - | - | ✓ | - |
| DMDc | ✓ | - | ✓ | - | - |
| eDMDc | ✓ | - | ✓ | ✓ | - |
| LatentODE | - | - | - | ✓ | - |

Balanced Neural ODEs are a model order reduction method and therefore result in a compromise between reconstruction accuracy and dimensional reduction to lower the computational burden. As we showed in our experiments, in comparison to other state-of-the-art methods, B-NODEs can significantly lower the model order, while only marginally increasing the prediction error. Computational speed can be increased by a factor of 8 compared to SS-NODE, as the ODE solver requires significantly less ODE function evaluations. As B-NODE is GPU compatible, a speed-up of 100 compared to physical simulation models is possible for batch simulations. The B-NODE framework is versatile enough to accommodate nonlinear model order reduction as well as linear Koopman operator approximation, and additionally allows for uncertainty quantification. Compared to MOR methods, B-NODEs provide a well-automatable method as extensive hyperparameter tuning (e.g. bottleneck dimension or observable functions) can be avoided.

Thus, BNODE offers a robust and versatile approach for surrogating detailed simulation models, making it applicable across diverse domains such as energy and automotive. Its flexibility enables efficient handling of a wide range of simulation tasks, including optimization, control, and other computationally demanding scenarios.

## 7 CONCLUSION

We present *Balanced Neural ODEs (B-NODEs)*, balancing state reduction and reconstruction quality, a well-automatable method for global nonlinear model order reduction. For that, we combine VAEs with state space models to leverage VAE's information bottleneck to enable measurable dimensional reduction without à priori defined latent dimensions. Additionally, the VAE's sampling-based robustness complements in literature reported stability enhancements for Neural ODEs with stochastic noise. We are motivated by the need for fast surrogate models with time-varying inputs. To achieve this, we continuously propagate variational parameters through time with a Neural ODE (to our awareness, for the first time). We tested our method on academic and real-world cases and show promising results in terms of reconstruction quality, generalization, model order reduction, and reduction of simulation time. We also showed that the B-NODE framework is well suited to learn Koopman operator approximations to linearize simulation models.

ACKNOWLEDGMENTS

A large part of this work was carried out as Julius Aka's master thesis at Hamburg University of Technology within the company XRG Simulation GmbH, substantially supported by University of Augsburg. J.A. would like to thank XRG Simulation GmbH for supporting this project and the possibility to make its content public. J.A. would also like to thank Yi Zhang and Tobias Thummerer from University of Augsburg for their support for this research.

The continuation oft this research was partially funded by the ITEA 4 project OpenScaling (*Open standards for SCALable virtual engineerING and operation*) N°22013 under grant number 01IS23062H. See `https://openscaling.org/` for more information.

REPRODUCIBILITY STATEMENT

We provide the source code on `https://github.com/juliusaka/balanced-neural-odes`. In addition to the code necessary for implementing VAE, B-NODE and SS-NODE, we also provide the files that implement the first-principle models along with FMUs (Modelica Association, 2022), containing simulators of those. We also provide the configuration files, that were used for the data generation process and training of all models.

Apart from this, we provided detailed explanation of B-NODEs in this paper, and included more details on sampling of first-principle models for dataset generation, data standardization, approximation of the data likelihood, and most important training settings in the appendix.

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

## A    APPENDIX

### A.1    DATA STANDARDIZATION

In this work, all values that are fed to the neural network models are standardized. This is especially of importance, as physical quantities might have very different order of quantities (e.g. a pressure at around $1 \times 10^5$ Pa compared to a temperature at around $50\,°$C). Also for loss functions, standardized values are used to avoid adapting weighting factors to the different orders of magnitudes. In general, the not-standardized values are used only for graphical presentation of results.

To standardize the data of a data set, mean $\boldsymbol{\mu}$ and variance $\boldsymbol{\sigma}$ are calculated for each variable. Then we use the following transformation for each axis $x$ (Prince (2023), C.2.4):

$$x_{\text{stand.}} = \frac{x - \boldsymbol{\mu}}{\boldsymbol{\sigma}} \tag{20}$$

To ensure the same standardization for every data sample, the mean and variance values have to be saved. Because of that, we introduce a *Standardization Activation Layer*. If not previously initialized, with the first of batch of data, mean and variance are calculated and saved as untrainable parameter in the *PyTorch*-Model. It can be used as a normal layer within the model, and has also the ability to inverse standardize - i.e.

$$x = \boldsymbol{\sigma} \cdot x_{\text{stand.}} + \boldsymbol{\mu}. \tag{21}$$

Opposed to batch-normalization (Prince (2023), 11.4), where the statistics $\boldsymbol{\mu}$ and $\boldsymbol{\sigma}$ are calculated for each batch independently, the statistics are constant.

Physical model parameters are standardized along each feature axis. All other time-varying values are standardized along the feature axis and the time axis.

For presentation and calculation of *summary statistics in this publication*, we chose to normalize by mean values:

$$x_{\text{norm.}} = \frac{x}{\boldsymbol{\mu}} \tag{22}$$

### A.2    STATE SPACE NEURAL ODE

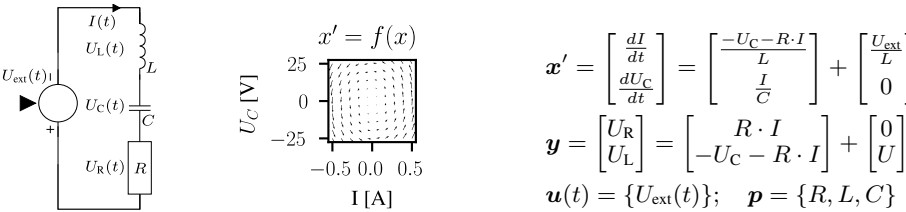

Figure A.1: The series resonant circuit with an voltage source as external input as an example of a dynamical system. We show its vector field (for no input) and its state space model.

#### A.2.1    TRAINING

Given a dataset $\boldsymbol{x}^{\mathcal{D}} = [\boldsymbol{x}_{0:T}, \boldsymbol{y}_{0:T}, \boldsymbol{u}_{0:T}, \boldsymbol{p}]$, the State Space Neural ODE is trained by minimizing the error of the predictions $[\hat{\boldsymbol{x}}(t), \hat{\boldsymbol{y}}(t)]$

$$\phi_{\text{NODE}}, \phi_{\text{Out}} = \underset{\phi_{\text{NODE}}, \phi_{\text{Out}}}{\text{argmin}} \, \text{MSE}(\hat{\boldsymbol{x}}_{0:T}, \boldsymbol{x}_{0:T}) + \text{MSE}(\hat{\boldsymbol{y}}_{0:T}, \boldsymbol{y}_{0:T}). \tag{23}$$

### A.3    VARIATIONAL AUTOENCODERS FOR DIMENSIONAL REDUCTION

#### A.3.1    ELBO DERIVATION

We could learn a model $p(\boldsymbol{x}|\boldsymbol{z})$ if we would know the latent variable $\boldsymbol{z}$. But we do not know $\boldsymbol{z}$. To overcome that, we would require the "true" model $p(\boldsymbol{z}|\boldsymbol{x})$ that represents the so called *posterior probability*[4] $\Pr(\boldsymbol{z}|\boldsymbol{x})$. But we do not have that model either. The learning problem is *intractable*.

---

[4]To explain why the term *posterior probability* is used for $\Pr(\boldsymbol{z}|\boldsymbol{x})$: We took the assumption that $\Pr(\boldsymbol{z})$ is the known prior probability. In terms of Bayes' Rule, after observing the generation process $\Pr(\boldsymbol{x}|\boldsymbol{z})$ we can

To make the learning problem tractable, a posterior model $q(\boldsymbol{z}|\boldsymbol{x})$ that approximates the true posterior probability model $p(\boldsymbol{z}|\boldsymbol{x})$ is introduced:

$$q(\boldsymbol{z}|\boldsymbol{x}) \approx p(\boldsymbol{z}|\boldsymbol{x}). \tag{24}$$

The model $q(\boldsymbol{z}|\boldsymbol{x})$ is called the *encoder* or *recognition model* (Kingma & Welling, 2019, Chapter 2.1). This approximation is the *variational approximation* that gives the VAE its name. (Prince, 2023, Chapter 17.5).
With this approximation, the latent variable model can be trained using data samples $\boldsymbol{x}$.

The following can be derived by using Jensen's inequality (as in (Prince, 2023, chapter 17.3.33)), but we choose a derivation by (Kingma & Welling, 2019, chapter 2.2) that avoids this.

To obtain the latent variable model, express the log-likelihood $\log p(\boldsymbol{x})$ of the data as an expectation over the posterior probability $q(\boldsymbol{z}|\boldsymbol{x})$

$$\log p(\boldsymbol{x}) = \mathbb{E}_{q(\boldsymbol{z}|\boldsymbol{x})} \log \left[ p(\boldsymbol{x}) \right]. \tag{25}$$

By applying the the product rule

$$\log p(\boldsymbol{x}) = \mathbb{E}_{q(\boldsymbol{z}|\boldsymbol{x})} \log \left[ \frac{p(\boldsymbol{x}, \boldsymbol{z})}{p(\boldsymbol{z}|\boldsymbol{x})} \right],$$

expanding with $\frac{q(\boldsymbol{z}|\boldsymbol{x})}{q(\boldsymbol{z}|\boldsymbol{x})}$

$$= \mathbb{E}_{q(\boldsymbol{z}|\boldsymbol{x})} \log \left[ \frac{p(\boldsymbol{x}, \boldsymbol{z})}{q(\boldsymbol{z}|\boldsymbol{x})} \frac{q(\boldsymbol{z}|\boldsymbol{x})}{p(\boldsymbol{z}|\boldsymbol{x})} \right],$$

splitting the integrand of the expectation

$$= \mathbb{E}_{q(\boldsymbol{z}|\boldsymbol{x})} \log \left[ \frac{p(\boldsymbol{x}, \boldsymbol{z})}{q(\boldsymbol{z}|\boldsymbol{x})} \right] + \mathbb{E}_{q(\boldsymbol{z}|\boldsymbol{x})} \log \left[ \frac{q(\boldsymbol{z}|\boldsymbol{x})}{p(\boldsymbol{z}|\boldsymbol{x})} \right]$$

and applying the product rule for the first term, we can derive

$$\log p(\boldsymbol{x}) = \underbrace{\mathbb{E}_{q(\boldsymbol{z}|\boldsymbol{x})} \log \left[ \frac{p(\boldsymbol{x}|\boldsymbol{z})}{q(\boldsymbol{z}|\boldsymbol{x})} p(\boldsymbol{z}) \right]}_{=\mathcal{G}(\text{ELBO})} + \underbrace{\mathbb{E}_{q(\boldsymbol{z}|\boldsymbol{x})} \log \left[ \frac{q(\boldsymbol{z}|\boldsymbol{x})}{p(\boldsymbol{z}|\boldsymbol{x})} \right]}_{=D_{\text{KL}}(q(\boldsymbol{z}|\boldsymbol{x})||p(\boldsymbol{z}|\boldsymbol{x}))}. \tag{26}$$

The KL-Divergence in the second term is always $\geq 0$. Therefore, the first term is a lower bound to the log-likelihood of the data. It is called the *evidence lower bound* (ELBO). (Kingma & Welling, 2019, Chapter 2.2) With the chosen variational approximation, it is possible to compute all parts of the ELBO. Before showing that, we note that the KL-Divergence of the second term determines two distances (Kingma & Welling, 2019, Chapter 2.2):

- The KL-Divergence between the approximate posterior and the true posterior,
- The gap between the ELBO and the marginal likelihood $\log p(\boldsymbol{x})$. This is called the *tightness* of the bound.

It is now possible to train a variational autoencoder by maximizing the ELBO $\mathcal{G}$. Maximizing ELBO will concurrently optimize two objectives (Kingma & Welling, 2019, Chapter 2.2.1):

- It will improve the prediction quality $\log p(\boldsymbol{x})$ of the generative model $p(\boldsymbol{x}|\boldsymbol{z})$.
- It will minimize the error of approximating the true posterior distribution $p(\boldsymbol{z}|\boldsymbol{x})$ through $q(\boldsymbol{z}|\boldsymbol{x})$, meaning that the encoder will get better.

The KL-Divergence in (3) for the comparison with the assumed prior $p(\boldsymbol{z}) = \mathcal{N}(\boldsymbol{z}; \boldsymbol{0}, \boldsymbol{I})$ can be solved analytically to (Odaibo, 2019)

$$D_{\text{KL}}\big(q_{\phi_{\text{en}}}(\boldsymbol{z}|\boldsymbol{x}^{\mathcal{D}})||p(\boldsymbol{z})\big) = \sum_{j=1}^{n_{\boldsymbol{z}}} -\frac{1}{2}\Big[ 1 + \log(\sigma_j^2) - \sigma_j^2 - \mu_j^2 \Big] \geq 0, \tag{27}$$

infer the posterior probability $\Pr(\boldsymbol{z}|\boldsymbol{x})$. It's a bit cumbersome to call this posterior probability. Even though this would be the first thing needed for the training of a VAE, but in terms of the VAE theory, the starting point is that the data generation process $\Pr(\boldsymbol{x}|\boldsymbol{z})$ can be traced back to a prescribed prior probability $\Pr(\boldsymbol{z})$.

with $n^z$ being the chosen dimensionality of the latent variable.

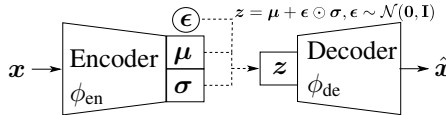

Figure A.2: Scheme of a Variational Autoencoder

### A.3.2 EXPERIMENT DETAILS

**Dataset**

The dataset was generated by random sampling of parameters of a physical simulation model with 14 parameters. The VAE model itself has no concept of sequences, therefore the sequences for 5 temperature trajectories formed one sample (i.e. 5 sequences $x$ 337 timesteps per sequence = 1685 datapoints per sample). We therefore re-implemented the model described in (Aka et al., 2023).

- parameter values: determined from building data
- parameter ranges: $\frac{1}{5}$ to $5$ times the default value
- timestep: $1800\,\mathrm{s}$
- simulation length: $7\,\mathrm{d}$
- number of timesteps: $337$
- number of samples: 2048 (12% test, 12% validate, 76% train)

**Experiment Details**

- Network Structure:
    - number of linear layers: 4
    - hidden dimension: 2048
    - latent space dimension: 512
    - activation function: ReLU

- Training Parameters:
    - beta: $1 \times 10^{-3}$
    - max epochs: $1 \times 10^{4}$
    - learning-rate: $1 \times 10^{-3}$
    - weight decay: $1 \times 10^{-4}$
    - batch size: $256$
    - early stopping patience: 300
    - early stopping threshold: 0.005
    - early stopping mode: relative
    - number of reparametrization passes for train and test: 1
    - automatic mixed precision: off
    - parameter to decoder: on (PELS-VAE), off (VAE)

- Code File: `.\networks\pels_vae_linear\vae_train_test.py`

### A.3.3 ADDITIONAL FIGURES

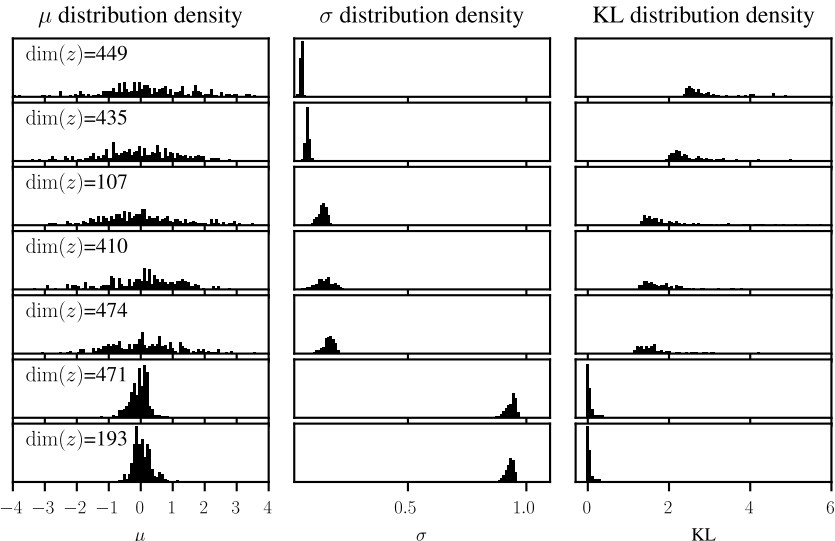

Figure A.3: Density distributions of $D_{\text{KL}}, \boldsymbol{\mu}, \boldsymbol{\sigma}$ for the first 7 latent space dimensions for the complete test set ordered by their mean KL divergence. Plot for the $\beta$-VAE with $\beta = 1 \times 10^{-3}$.The model was trained with a data set of the office model

To verify that the first 5 dimension encode most information, we ordered the latent dimensions of a trained VAE by their average $D_{\text{KL}}$, masked all latent channels to 0 and consecutively lifted that mask while calculating for each configuration the MSE on the dataset (Figure A.4a). As anticipated, the first 5 latent channels reduce the error by the most ($0.85 \mapsto 0.09$) while the last 507 dimensions only achieve marginal improvements ($0.09 \mapsto 0.04$). Notably, the improvements are monotonously decreasing. This motivates to use the mean $D_{\text{KL}}$ per latent channel as a measure for active latent dimensions, and we To find a threshold value, we tested a variety of values in Figure A.4b and choose for counting a value of $D_{\text{KL}}(z) > 0.1$ (Figure A.4b).

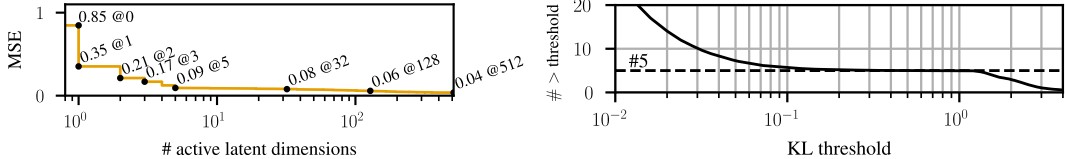

(a) Experiment: MSE on dataset for increasing number of active latent dimensions by lifting a mask on the $D_{\text{KL}}$-ordered latent dimensions.

(b) Number of latent dimensions that have a KL divergence greater than a certain threshold, averaged over one dataset.

Figure A.4: Insights in the information transmission through the latent space. Results of a $\beta$-VAE ($\beta = 1 \times 10^{-3}$) with a latent space of $n_{\boldsymbol{z}} = 512$ trained on a dataset of a thermal network model, generated by random sampling of 14 simulation model parameters (details in A.3.2).

### A.4 BALANCED NEURAL ODEs

#### A.4.1 B-NODE ELBO DERIVATION

In the following we expand the ELBO to our use case. As provided by Kingma & Welling (2014), the ELBO is

$$\mathcal{G} = \underbrace{\mathbb{E}_{q(\boldsymbol{z}|\boldsymbol{x}^{\mathcal{D}})} \log\left[p(\boldsymbol{x}^{\mathcal{D}}|\boldsymbol{z})\right]}_{\text{Reconstruction Accuracy}} - \underbrace{D_{\text{KL}}\left[q(\boldsymbol{z}|\boldsymbol{x}^{\mathcal{D}})||p(\boldsymbol{z})\right]}_{\text{Information Bottleneck}}. \tag{28}$$

First, we collect the governing equations of the model and adapt the way they are expressed ((5), (5a), (5b), (5ba), (5), (6)). The approximated posterior (inference model) is

$$q(\boldsymbol{z}|\boldsymbol{x}^{\mathcal{D}}) = q(\boldsymbol{x}_{0:T}^z, \boldsymbol{u}_{0:T}^z, \boldsymbol{p}^z|\boldsymbol{x}_{0:T}, \boldsymbol{u}_{0:T}, \boldsymbol{p}) = q(\boldsymbol{p}^z|\boldsymbol{p})q(\boldsymbol{u}_{0:T}^z|\boldsymbol{u}_{0:T})q(\boldsymbol{x}_{0:T}^z|\boldsymbol{x}_{0:T}, \boldsymbol{u}_{0:T}^z, \boldsymbol{p}^z),$$

$$= \underbrace{\prod_{i=0}^{T} q(\boldsymbol{u}_i^z|\boldsymbol{u}_i)}_{\text{Control Encoder}} \underbrace{q(\boldsymbol{p}^z|\boldsymbol{p})}_{\text{Parameter Encoder}} \underbrace{q(\boldsymbol{x}_0^z|\boldsymbol{x}_0)}_{\text{Initial State Observer}} \underbrace{\prod_{i=0}^{T-1} q(\boldsymbol{x}_{i+1}^z|\boldsymbol{x}_i^z, \boldsymbol{u}_i^z, \boldsymbol{p}^z)}_{\text{Simulator}}$$

$$= \underbrace{q(\boldsymbol{u}_{0:T}^z, \boldsymbol{p}^z|\boldsymbol{u}_{0:T}, \boldsymbol{p})}_{\text{independent sampling possible}} \underbrace{q(\boldsymbol{x}_{0:T}^z|\boldsymbol{x}_0, \boldsymbol{u}_{0:T}^z, \boldsymbol{p}^z)}_{\text{dependent on previous sampling}}, \tag{29a}$$

the generative model is

$$p(\boldsymbol{x}^{\mathcal{D}}|\boldsymbol{z}) = p(\boldsymbol{x}_{0:T}, \boldsymbol{y}_{0:T}|\boldsymbol{x}_{0:T}^z, \boldsymbol{u}_{0:T}^z, \boldsymbol{p}^z) = \prod_{i=0}^{T} p(\boldsymbol{x}_i, \boldsymbol{y}_i|\boldsymbol{x}_i^z, \boldsymbol{u}_i^z, \boldsymbol{p}^z), \tag{29b}$$

and we prescribed the prior as

$$p(\boldsymbol{z}) = p(\boldsymbol{p}^z)p(\boldsymbol{x}_{0:T}^z)p(\boldsymbol{u}_{0:T}^z). \tag{29c}$$

When inserting these terms in the ELBO, the learning problem is intractable, as the probability density functions are parameterized by outputs of neural networks that cannot be solved analytically. Therefore, sampling from the posterior distribution $q(\boldsymbol{z}|\boldsymbol{x}^{\mathcal{D}})$ in conjunction with the reparameterization trick is necessary, whenever latent variables are propagated from deep neural network to another. The propagation of the latent state distribution by the Neural ODE is an exception for this, this propagation of probability distributions is implemented tractably and state reduction ensured by the sampling of inputs (13).

First, we derive the data likelihood part of the ELBO,

$$\mathbb{E}_{q(\boldsymbol{z}|\boldsymbol{x}^{\mathcal{D}})} \log\left[p(\boldsymbol{x}^{\mathcal{D}}|\boldsymbol{z})\right] = \mathbb{E}_{q(\boldsymbol{u}_{0:T}^z, \boldsymbol{p}^z|\boldsymbol{u}_{0:T}, \boldsymbol{p})q(\boldsymbol{x}_{0:T}^z|\boldsymbol{x}_0, \boldsymbol{u}_{0:T}^z, \boldsymbol{p}^z)} \log\left[p(\boldsymbol{x}_{0:T}, \boldsymbol{y}_{0:T}|\boldsymbol{x}_{0:T}^z, \boldsymbol{u}_{0:T}^z, \boldsymbol{p}^z)\right],$$

Considering the sampling dependencies,

$$= \mathbb{E}_{q(\boldsymbol{u}_{0:T}^z, \boldsymbol{p}^z|\boldsymbol{u}_{0:T}, \boldsymbol{p})}\left[\mathbb{E}_{q(\boldsymbol{x}_{0:T}^z|\boldsymbol{x}_0, \boldsymbol{u}_{0:T}^z, \boldsymbol{p}^z)} \log\left[p(\boldsymbol{x}_{0:T}, \boldsymbol{y}_{0:T}|\boldsymbol{x}_{0:T}^z, \boldsymbol{u}_{0:T}^z, \boldsymbol{p}^z)\right]\right],$$

Using that $p(\boldsymbol{x}^{\mathcal{D}}|\boldsymbol{z})$ can be factorized,

$$= \mathbb{E}_{q(\boldsymbol{u}_{0:T}^z, \boldsymbol{p}^z|\boldsymbol{u}_{0:T}, \boldsymbol{p})}\left[\mathbb{E}_{q(\boldsymbol{x}_{0:T}^z|\boldsymbol{x}_0, \boldsymbol{u}_{0:T}^z, \boldsymbol{p}^z)}\left[\sum_{i=0}^{T} \log p(\boldsymbol{x}_i, \boldsymbol{y}_i|\boldsymbol{x}_i^z, \boldsymbol{u}_i^z, \boldsymbol{p}^z)\right]\right]. \tag{30}$$

After this, we derive the KL divergence part. By using the derivation in A.4.4, we can see that the KL divergence splits up in separate terms for different random variables. Because of this,

$$D_{\text{KL}}\left[q(\boldsymbol{z}|\boldsymbol{x}^{\mathcal{D}})||p(\boldsymbol{z})\right] = D_{\text{KL}}\left[q(\boldsymbol{p}^z|\boldsymbol{p})q(\boldsymbol{u}_{0:T}^z|\boldsymbol{u}_{0:T})q(\boldsymbol{x}_{0:T}^z|\boldsymbol{x}_0, \boldsymbol{u}_{0:T}^z, \boldsymbol{p}^z)||p(\boldsymbol{p}^z)p(\boldsymbol{u}_{0:T}^z)p(\boldsymbol{x}_{0:T}^z)\right].$$

As $q(\boldsymbol{x}_{0:T}^z|\boldsymbol{x}_0, \boldsymbol{u}_{0:T}^z, \boldsymbol{p}^z)$ depends on previous sampling, we cam write this as,

$$= D_{\text{KL}}\left[q(\boldsymbol{p}^z|\boldsymbol{p})||p(\boldsymbol{p}^z)\right] + D_{\text{KL}}\left[q(\boldsymbol{u}_{0:T}^z|\boldsymbol{u}_{0:T})||p(\boldsymbol{u}_{0:T}^z)\right]$$
$$+ \mathbb{E}_{q(\boldsymbol{u}_{0:T}^z, \boldsymbol{p}^z|\boldsymbol{u}_{0:T}, \boldsymbol{p})}\left(D_{\text{KL}}\left[q(\boldsymbol{x}_{0:T}^z|\boldsymbol{x}_0, \boldsymbol{u}_{0:T}^z, \boldsymbol{p}^z)||p(\boldsymbol{x}_{0:T}^z)\right]\right). \tag{31}$$

Each variable in the KL divergence defined here expands likewise as a sum of KL divergence terms. Because we prescribe (parameterized) Gaussians, the KL divergence can be solved analytically. The model is then trained as

$$\phi_{\text{en,p}}, \phi_{\text{en,u}}, \phi_{\text{en,x}}, \phi_{\text{LNODE}}, \phi_{\text{de}} = \underset{\phi}{\text{argmax}} \quad \mathcal{G}\big(\phi_{\text{en,p}}, \phi_{\text{en,u}}, \phi_{\text{en,x}}, \phi_{\text{LNODE}}, \phi_{\text{de}}, \boldsymbol{x}_{0:T}, \boldsymbol{u}_{0:T}, \boldsymbol{p}\big) \quad (32)$$

*Remark:* Typically, as proposed by Kingma & Welling (2014), we sample each distribution only one time during training and train with mini-batches. Although we elaborated here on the sampling dependencies, it is not necessary to obey to this when always sampling once.

*Remark:* Opposed to Girin et al. (2020), we do not condition the true posterior probability density of the generative model for the latent states at time $t_k$ on the previous probability densities (for $\boldsymbol{x}_{i-1}^z, \boldsymbol{u}_{i-1}^z, \boldsymbol{p}^z$), but prescribe the latent space in general to follow a Gaussian distribution with mean zero and variance of one ($\mathcal{N}(\boldsymbol{0}, \mathbb{I})$) at all discrete times $t \in [0, T]$. This means that we do not condition the generative model in equation 4.3 in their review on anything, and means as well that all conditional probabilities evolving an advance in time are collected in the inference model.

### A.4.2 NORMALIZATION OF ELBO TERMS

We normalize the reconstruction term with $\frac{1}{2}$ to account for the summation of the state reconstruction and output reconstruction MSE,

$$\mathcal{G}_{\text{R}} \approx -\frac{1}{2}\big(\text{MSE}(\hat{\boldsymbol{x}}, \boldsymbol{x}) + \text{MSE}(\hat{\boldsymbol{y}}, \boldsymbol{y})\big). \quad (33)$$

Let $D_{\text{KL},\square}$ be the KL divergence for $\square \in [\boldsymbol{p}, \boldsymbol{u}_k, \boldsymbol{x}_k], k \in [0, T]$. We calculate the Kullback-Leibler part of the ELBO as

$$\mathcal{L}_{\text{KL}} = \frac{1}{n_{\boldsymbol{p}} + n_{\boldsymbol{u}} + n_{\boldsymbol{x}}}\Big(D_{\text{KL},\boldsymbol{p}} + \frac{1}{T}\sum_{k=0}^{T} D_{\text{KL},\boldsymbol{u}_k} + \frac{1}{T}\sum_{k=0}^{T} D_{\text{KL},\boldsymbol{x}_k}\Big) \quad (34)$$

Time-varying variables are normalized by the number of time steps $T$. Furthermore, we choose to normalize the components by their dimensions in physics space $n_\square$ in order to being able to use the same $\beta$ for different use cases.

### A.4.3 APPROXIMATION OF DATA LOG-LIKELIHOOD

Optimizing the log-likelihood of $p(\boldsymbol{x}|\boldsymbol{z})$ can be represented by (Prince, 2023, 5.3)

$$\underset{\phi_{\text{de}}}{\text{argmax}} \quad \prod_{i=0}^{T} p(\boldsymbol{x}_i, \boldsymbol{y}_i | \boldsymbol{x}_i^z, \boldsymbol{u}_i^z, \boldsymbol{p}^z) = \underset{\phi_{\text{de}}}{\text{argmin}} \quad \sum_{i=0}^{T} \text{MSE}([\hat{\boldsymbol{x}}_i, \hat{\boldsymbol{y}}_i], [\boldsymbol{x}_i, \boldsymbol{y}_i]), \quad (35)$$

where $[\hat{\boldsymbol{x}}_i, \hat{\boldsymbol{y}}_i]$ is the output of (15) and $p$ is assumed to be a Gaussian with diagonal, fixed covariance.

### A.4.4 KL-DIVERGENCE OF MULTIPLE VARIABLES

The KL-Divergence between two distribution $p(z), q(z)$ is defined as

$$D_{\text{KL}}(q(z)||p(z)) = \int_z q(z) \log\left[\frac{p(z)}{q(z)}\right] dz$$

For the product of two distributions of different random variables $z_1, z_2$ this definition yields:

$$D_{\text{KL}}(q(z_1)q(z_2)||p(z_1)p(z_2))$$
$$= \int q(z_1)q(z_2) \log\left[\frac{p(z_1)p(z_2)}{q(z_1)q(z_2)}\right] d\boldsymbol{z}$$
$$= \int_{z_2}\int_{z_1} q(z_1)q(z_2) \log\left[\frac{p(z_1)p(z_2)}{q(z_1)q(z_2)}\right] dz_1 dz_2$$

Using logarithm laws

$$= \int_{z_2} \int_{z_1} q(z_1)q(z_2) \left( \log \left[ \frac{p(z_1)}{q(z_1)} \right] + \log \left[ \frac{p(z_2)}{q(z_2)} \right] \right) dz_1 dz_2$$

Splitting the integral

$$= \int_{z_2} \int_{z_1} q(z_1)q(z_2) \log \left[ \frac{p(z_1)}{q(z_1)} \right] dz_1 dz_2 + \int_{z_2} \int_{z_1} q(z_1)q(z_2) \log \left[ \frac{p(z_2)}{q(z_2)} \right] dz_1 dz_2$$

Moving the integration symbols and sorting the terms

$$= \int_{z_2} q(z_2) \int_{z_1} q(z_1) \log \left[ \frac{p(z_1)}{q(z_1)} \right] dz_1 dz_2 + \int_{z_1} q(z_1) \int_{z_2} q(z_2) \log \left[ \frac{p(z_2)}{q(z_2)} \right] dz_2 dz_1$$

Inserting the definition of KL-Divergence again

$$= \int_{z_2} q(z_2) D_{\text{KL}} \left( q(z_1) || p(z_1) \right) dz_2 + \int_{z_1} q(z_1) D_{\text{KL}} \left( q(z_2) || p(z_2) \right) dz_1$$

using that probability density functions always integrate to 1, we see that

$$= D_{\text{KL}} \left( q(z_1) || p(z_1) \right) + D_{\text{KL}} \left( q(z_2) || p(z_2) \right). \tag{36}$$

Hence we follow that the KL-Divergence of two distributions on different random variables always yields a sum of KL-Divergence terms for the single variable types.

## A.5 EXPERIMENTS

### A.5.1 DATASET GENERATION FROM PHYSICAL SIMULATION MODELS

The surrogate models shall replace the physical models based on first principles. The way we choose to train these surrogates is by imitating the data $[\boldsymbol{x}_k, \boldsymbol{y}_k]$ at discretization points $k = 0 \ldots T$ given $[\boldsymbol{x}_0, \boldsymbol{p}, \boldsymbol{u}_k]$. Therefore, we need data of the input-output pairs:

$$(\boldsymbol{x}_0, \boldsymbol{p}, \boldsymbol{u}_{0:T}) \mapsto (\boldsymbol{x}_{1:T}, \boldsymbol{y}_{0:T})$$

We choose the control input to be constant in the discretization intervals:

$$\boldsymbol{u}(t) = \boldsymbol{u}_k = \text{const.} \quad \forall t \in [t_k, t_{k+1}]$$

From experience, uniform random sampling proved to allow good learning for values that are constant over a simulation, which is why we sample the parameters and initial values like that:

$$\boldsymbol{x}_0 \sim \mathcal{U}(\boldsymbol{x}_{0,-}, \boldsymbol{x}_{0,+}) \tag{37}$$
$$\boldsymbol{p} \sim \mathcal{U}(\boldsymbol{p}_-, \boldsymbol{p}_+) \tag{38}$$

with $\mathcal{U}$ denoting independent random sampling, and $\square_-, \square_+$ denoting the user-specified upper and lower values for sampling. These values should be chosen sufficiently larger than the area of interest, such that extrapolation is avoided. However, note that in general

$$\min \boldsymbol{x}_{1:T} \neq \boldsymbol{x}_{0,-} \quad \text{and} \quad \max \boldsymbol{x}_{1:T} \neq \boldsymbol{x}_{0,+}$$

as the states can not be bounded à priori.
Sampling the initial state can be seen as a way to increase the expressiveness of the training data to special regions of states space, which are otherwise rarely seen or unseen during training. When generating training data, it should be ensured that required initial states for the surrogate purpose are included and well present.
However, variation of initial states for time-stepper methods in typical model regions can always be included with the following. Recall that time-stepper methods build up vector fields:

$$(\boldsymbol{x})' = f(\boldsymbol{x}, \boldsymbol{p}, \boldsymbol{u})$$

This makes it possible to enter the training at arbitrary locations within the time series sequences. This works also in latent space but requires a transformation of the physical initial state $\boldsymbol{x}_0 \mapsto \boldsymbol{x}_0^z$ to latent space. We can always adapt the initial states during training to typical state vectors within the specific physical model use case, by choosing a training sequence length $\tau \in \mathcal{N}$ with $\tau < T$ with

$T \in \mathcal{N}$ being the sequence length of the training data and generate the samples in the following from it:

$$\tilde{\boldsymbol{x}}_0, \boldsymbol{p}, \tilde{\boldsymbol{x}}_{1:\tau}, \tilde{\boldsymbol{u}}_{0:\tau}, \tilde{\boldsymbol{y}}_{0:\tau} = \boldsymbol{x}_j, \boldsymbol{p}, \boldsymbol{x}_{j+1:j+\tau}, \boldsymbol{u}_{j:j+\tau}, \boldsymbol{y}_{j:j+\tau} \qquad \forall j \in [0, 1, \ldots T - \tau] \qquad (39)$$

Note that this increases the number of training samples by large magnitude, i.e. with $T = 500$ and $\tau = 50$, we get $T - \tau = 450$ new samples per sequence. Therefore in a software implementation, the generated samples should not be saved on file, but returned on the fly from the original time series sequences to keep memory requirements low (at the expense of computational cost).

For the time-dependent inputs, simple random sampling does not work. With necessarily small discretization interval $\Delta t$, random sampling would lead to pure noise around a mean-value $\bar{\boldsymbol{u}}_{0:T}$, such that all training samples would cycle around the same position in state space. Therefore, in the following, we introduce a method called *randomly clipped random sampling of controls with offset and cubic splines (RROCS)*. Pseudo code of the algorithm is depicted in 1. The main idea is to ensure a smooth control input within the specified bounds through the use of cubic splines, while randomly assigning an offset and an amplitude such that the control space is filled meaningfully.

---

**Algorithm 1** randomly clipped random sampling of controls with offset and cubic splines (RROCS) algorithm

---

**for** $i$ in $n_{samples}$ **do**
    **for** $j$ in $n_{\boldsymbol{u}}$ **do**
        $t_{\text{sampled}} \leftarrow$ random sequence with min. and max. frequency
        $u_{\text{sampled}} \leftarrow$ sequence with random offset and amplitude$(u_-, u_+)$
        $u_{\text{spline}} \leftarrow$ Cubic Spline$(t_{\text{sampled}}, u_{\text{sampled}}, \boldsymbol{t}_{0:T})$         ▷ Ensure smoothness
        $u_{\text{norm}} \leftarrow$ Normalize $u_{\text{spline}}$ to be in [0,1]
        $b, \Delta \leftarrow$ random$(u_-, u_+)$         ▷ Push controls to different regions
        $b, \Delta \leftarrow$ clip such that u within bounds$(b, \Delta, u_-, u_+)$         ▷ Ensure constraints on input
        $u_{i,j,0:T} \leftarrow b + u_{\text{norm}} \cdot \Delta$
    **end for**
**end for**

---

An example for control inputs generated by this algorithm is given in Figure A.5

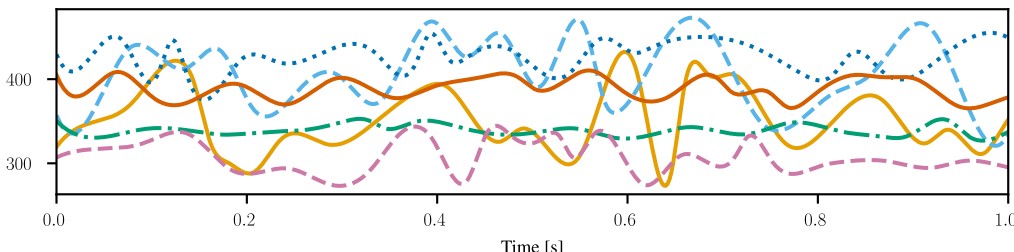

Figure A.5: Six example outputs of the *RROCS*-algorithm for the a control input `temperature_K_a` with $\boldsymbol{u}_- = 273.15, \boldsymbol{u}_+ = 473.15$.

If parameters or initial states are not sampled, they are set to a default value $\square_\diamond$.

## A.5.2 STRATIFIED HEAT FLOW MODEL (SHF) MODEL

The stratified heat flow models presented in Figure A.6 are introduced because they are a simplified version of discretized (energy) transport phenomena that often arise in the system simulation context, e.g. fluid transport in pipes, discretized heat transfer in heat exchangers, discretized modeling of passenger cabins or battery packs. These kind of discretizations are often necessary for exact modeling of the different use-case, but often result in models that are computationally too expensive for tasks like optimization, calibration or controller tuning. We anticipate that depending on the specific implementation of the discretized segments correlations between some states can be found. For example, for the stratified heat flow model controlled at both ends, the segments in the middle of the model show probably very similar temperature.

**Dataset Generation:**

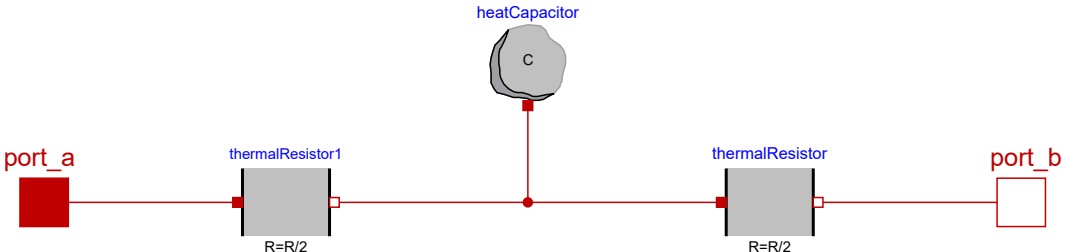

(a) Base R2C1 thermal network with parameters R and C for the total resistance and capacitance. The stratified heat flow model is derived by repeated connection of this base network, where each segment has the parameters $R_i = R/\texttt{nSeg}$ and $C_i = C/\texttt{nSeg}$ with nSeg being the number of discretization layers.

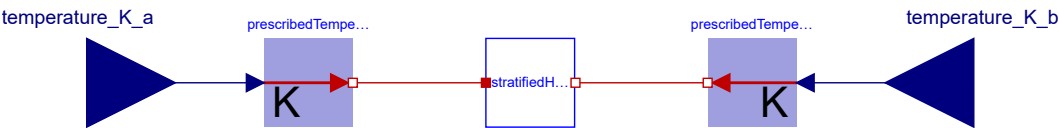

(b) Stratified heat flow model with parameters R and C for the total resistance and capacitance, and the control inputs temperature_K_a and temperature_K_b. The heat flow is discretized into $\texttt{nSeg} = 16$ segments.

Figure A.6: Stratified heat flow models created with Modelica (Association, 2023).

- number of samples: 1024
- training fraction: 76 %
- test fraction: 12 %
- validation fraction: 12 %
- Control Input Sampling Ranges:
  - temperature_K_a: [273.15, 473.15] K
  - temperature_K_b: [273.15, 473.15] K
- Solver Settings:
  - start time: $0\,\text{s}$
  - stop time: $1.2\,\text{s}$
  - output interval: $0.002\,\text{s}$
  - tolerance: $1 \times 10^{-6}$
  - sequence length: 501
  - the first $0.2\,\text{s}$ are not included in the datasets.
- 16 states
- 48 outputs

**Training:**

- Hyperparameters of all networks (state encoder, control encoder, parameter encoder, Latent ODE function, decoder):
  - 4 layers
  - hidden dimension of 128
  - Activation: ELU
  - latent space dimension for latent parameter, state and control space: 16
- Training:
  - $\beta$: different values
  - Adam Optimizer
  - learning rate: $1 \times 10^{-3}$
  - weight decay: $1 \times 10^{-5}$
  - main training
    * Batches per Epoch: 12
    * Phase 1:
      · Solver: rk4
      · training sequence length: 10
    * Phase 2:
      · Solver: rk4

· training sequence length: 250
· increase training sequence length over n batches: 100
· abort training sequence length increase after n stable epochs: 10
∗ Phase 3:
· Solver: rk4
· training sequence length: 400
· increase training sequence length over n batches: 100
· abort training sequence length increase after n stable epochs: 10

### A.5.3 POWER PLANT MODEL

The model is a thermal power plant model of the *ClaRa-Library* (Vojacek et al., 2023). Specifically, the `SteamCycle_01` model from the examples package of that library [5]. We refer to this model in the following as the *steam cycle* model.

It is a model of a $580\,\mathrm{MW}$ hard coal power plant with a single reheater, one high-pressure preheater

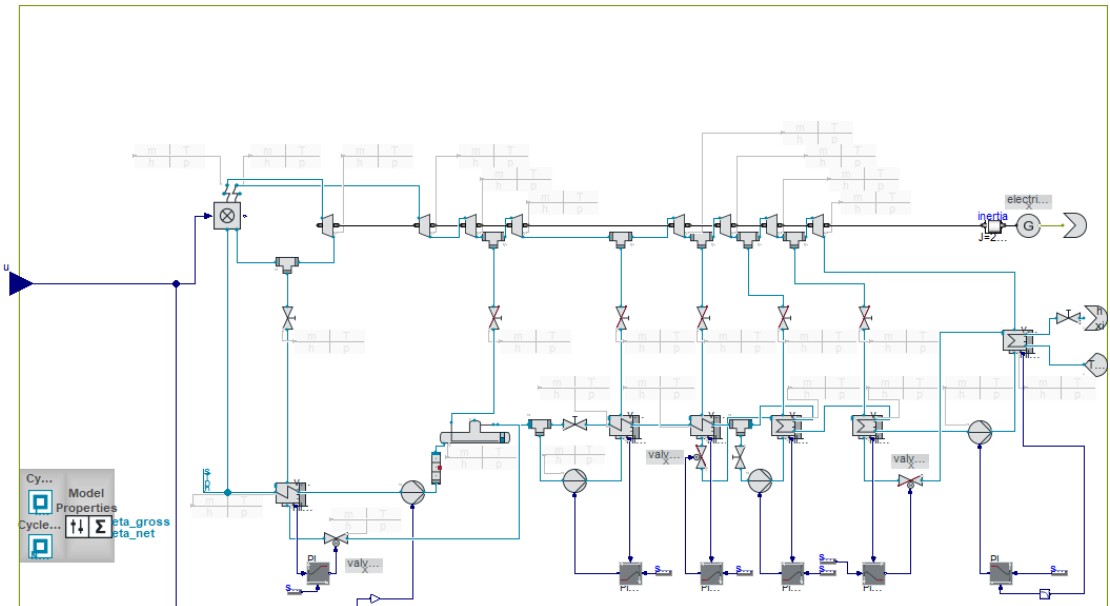

Figure A.7: Diagram view of the steam cycle model. (Vojacek et al., 2023)

and four low-pressure preheaters. To find initialization values for the states, a static-cycle for the same power plant is included and provides these values for the start of the simulation. The model has several internal PI-controllers that control the pressure-levels of the preheaters and condensers. The heating power and feedwater pump power is controlled by an input signal, the relative power. This is the only signal we considered for this application.

The system has 133 states, of that some were inactive, such that surrogate models are built for 102 states. It furthermore contains 34 nonlinear systems.

The example sets two $10\,\mathrm{s}$ load reductions as input signals over a total simulation time of $5000\,\mathrm{s}$. To make the model compatible with this work, an input connector was added and the initialization was modified to take the initial value from the input connector signal.

**Dataset Generation:**

- number of samples: 1024
- training fraction: $76\,\%$
- test fraction: $12\,\%$
- validation fraction: $12\,\%$
- Control Input Sampling Ranges:
    – `u`: [0.6, 1.1]

---

[5] `https://github.com/xrg-simulation/ClaRa-official` (Team)

- Solver Settings:
    - start time: $0\,\mathrm{s}$
    - stop time: $6000\,\mathrm{s}$
    - the first $1000\,\mathrm{s}$ were omitted to discard transient behavior from initialization
    - output interval: $5\,\mathrm{s}$
    - tolerance: $1 \times 10^{-6}$
    - sequence length: 2500
- 102 states (Prediction results only shown in **??**)
- 7 outputs (RMSE in this work for output prediction error)

**Training:**

- Hyperparameters of all networks (state encoder, control encoder, parameter encoder, Latent ODE function, decoder OR Neural ODE function, output network):
    - 4 layers
    - hidden dimension of 128
    - Activation: ELU
    - latent space dimension for latent parameter, state and control space: 128
- For Koopman approximation: latent control, latent state dimension: 1024
- Initialization: random (B-NODE, NODE), eigenvalues $< 0$ (B-NODE Koopman)
- Training:
    - $\beta$: different values
    - Adam Optimizer
    - learning rate: $1 \times 10^{-4}$ (B-NODE, B-NODE Koopman), $1 \times 10^{-5}$ (NODE)
    - weight decay: $1 \times 10^{-5}$, $1 \times 10^{-8}$ (B-NODE Koopman, NODE)
    - clip grad norm: 2.0
    - main training
        * Batches per Epoch: 12
        * Phase 1:
            · Solver: euler (rk4 B-NODE Koopman)
            · training sequence length: 5 (10 NODE)
            · break after loss of: 0.2 (0.05 NODE)
        * Phase 2:
            · Solver: rk4
            · training sequence length: 80
            · increase training sequence length over n batches: 6000 (1200 NODE)
            · abort training sequence length increase after n stable epochs: 1000
        * Phase 3:
            · Solver: rk4
            · training sequence length: 250
            · increase training sequence length over n batches: 1200
            · abort training sequence length increase after n stable epochs: 1000

### A.5.4 DETAILS FOR COMPARISON OF COMPUTATION TIME FOR POWER PLANT MODEL

For generation of data from the power plant model[6] (example 1), the model was exported as a co-simulation FMU with *CVODE* solver with Dymola [7] and simulated with FMPy[8]

- solving one system:
    - computation time: 16 seconds
    - hardware specifications:
        * 8-core processor with up to 4.9 GHz
    - technical setup:
        * solved using FMPy with per-time-step communication
- training data generation of 1024 samples:

---

[6]ClaRa official: `https://github.com/xrg-simulation/ClaRa-official`
[7]Dymola: `https://www.3ds.com/products/catia/dymola`
[8]FMPy: `https://github.com/CATIA-Systems/FMPy`

- – computation time: 720 seconds
- – hardware specifications:
  - ∗ 16-core processor with up to 5.7 GHz
  - ∗ 128 GB RAM
- – technical setup:
  - ∗ parallel processing using Dask[9]
  - ∗ simulations executed via FMPy with per-time-step communication
- simulation with surrogate models on GPU:
  - – model trained with dopri5 solver in the last period
  - – for adaptive step size control, the sample with the highest error determines step size control.
  - – hardware specifications:
    - ∗ Nvidia RTX4080 with 16 GB of GPU memory
  - – technical setup:
    - ∗ all samples are solved concurrently in a single batch, provided that sufficient GPU memory is available.

### A.5.5 KOOPMAN ANALYTIC EXAMPLE

We implemented (19) as Modelica model.

**Dataset Generation:**

- number of samples: 1024
- training fraction: 76 %
- test fraction: 12 %
- validation fraction: 12 %
- initial state sampling ranges:
  - – `x1`: [-50, 50]
  - – `x2`: [-50, 50]
- solver settings:
  - – start time: $0\,\mathrm{s}$
  - – stop time: $10\,\mathrm{s}$
  - – output interval: $0.1\,\mathrm{s}$
  - – tolerance: $1 \times 10^{-5}$
  - – sequence length: 99
- 2 states

**Training:**

- latent state dimension: 64
- Training:
  - – $\beta$: different values
  - – Adam Optimizer
  - – learning rate: $1 \times 10^{-3}$
  - – weight decay: $1 \times 10^{-5}$
  - – clip grad norm: 1.0
  - – main training
    - ∗ Batches per Epoch: 12
    - ∗ Phase 1:
      - · Solver: euler
      - · training sequence length: 5
      - · break after loss of: 0.2
    - ∗ Phase 2:
      - · Solver: rk4
      - · training sequence length: 30

---

[9]Dask: `https://www.dask.org/`

· increase training sequence length over n batches: 1200
· abort training sequence length increase after n stable epochs: 10
∗ Phase 3:
· Solver: rk4
· training sequence length: 80
· increase training sequence length over n batches: 1200
· abort training sequence length increase after n stable epochs: 10

### A.5.6  SMALL GRID MODEL

The small grid model describes a simplified 3 dimensional CFD simulation of air flow in a cubic grid consisting of 3x3x3 grid cells. The cube in the center is modeled as a solid mass. Two grid cells at the sides are connected to sources of prescribed temperature and mass flow, and a third cell is connected to an outflow port, such that a directed flow pattern evolves in the grid, cycling around the cube in either one or another direction (Figure A.8).

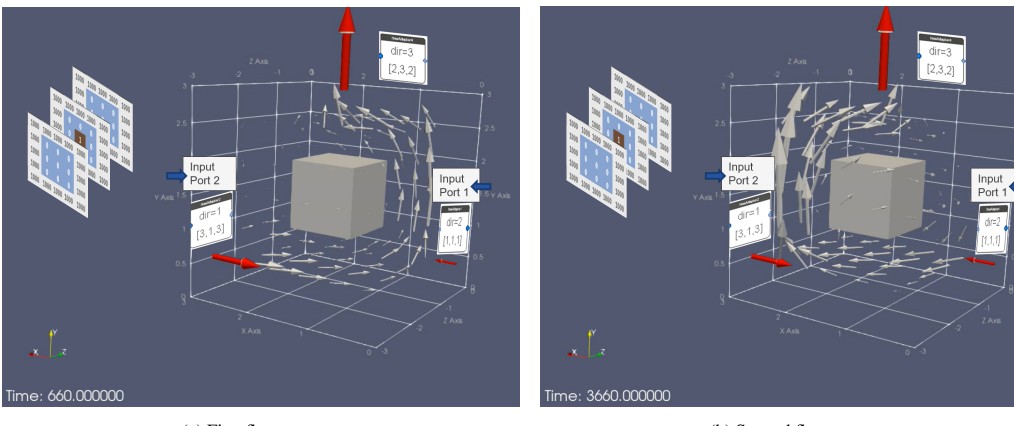

(a) First flow pattern.

(b) Second flow pattern.

Figure A.8: Visualization of simulation results of the small grid model.

The model is implemented using the Modelica *HumanComfort Library*[10]. We apply *RROCS*-sampling to generate a datset.

**Dataset Generation:**

- number of samples: 1024
- training fraction: 76 %
- test fraction: 12 %
- validation fraction: 12 %
- Control Input Sampling Ranges:
  - T_in: [273, 310] K
  - m_flow_in: [0.0, 1.0] $\mathrm{kg\,s^{-1}}$
  - T_in1: [273, 310] K
  - m_flow_in1: [0.0, 1.0] $\mathrm{kg\,s^{-1}}$
- Solver Settings:
  - start time: $0\,\mathrm{s}$
  - stop time: $1100\,\mathrm{s}$
  - output interval: $0.25\,\mathrm{s}$
  - tolerance: $1 \times 10^{-4}$
  - sequence length: 4001
  - the first $100\,\mathrm{s}$ are not included in the datasets.
- 100 states

---

[10]https://xrg-simulation.de/en/seiten/humancomfort-library

A FMU [11] of the model or the generated dataset is available on request.

### A.5.7 WATER HAMMER MODEL

The Waterhammer model consists of a pipe filled with water, whose ends are connected to a pressure and a mass flow source (Lenord, 2024). Through the pressure source, a pressure change can be applied, such that a pressure wave is generated that is reflected at the ends of the pipe. Evaporation effects of water are neglected. To model the spatial distribution of mass flow and pressure, a high discretization of the pipe's volume is necessary. We choose a discretization of 100 control volumes over pipe length of 10m. The Model is implemented using the *ClaRa* Library (Vojacek et al., 2023).

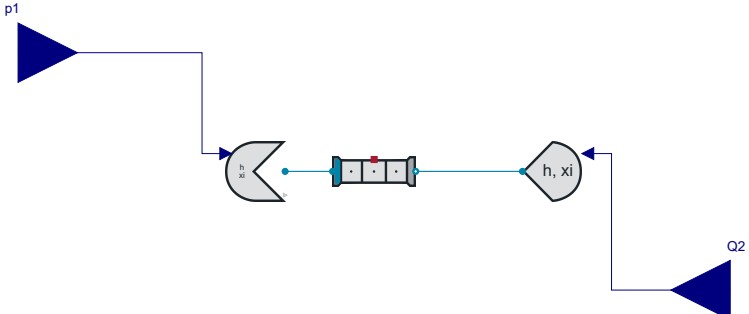

Figure A.9: Diagram view of the water hammer model.

**Dataset Generation:**

- number of samples: 1024
- training fraction: 76 %
- test fraction: 12 %
- validation fraction: 12 %
- Control Input Sampling Ranges:
    - sampling of a pressure ramp with parameters:
        * `p_start`: [5e5,20e5] Pa
        * `p_increase`: [1e4,80e5] Pa
        * `time_for_ramp`: [0.0001,0.2] s
    - constant mass flow rate between $0.001\,\mathrm{kg\,s^{-1}}$ to $0.1\,\mathrm{kg\,s^{-1}}$
- Solver Settings:
    - start time: $0\,\mathrm{s}$
    - stop time: $0.5\,\mathrm{s}$
    - output interval: $1 \times 10^{-3}\,\mathrm{s}$
    - tolerance: $1 \times 10^{-6}$
    - sequence length: 351
    - the first $0.15\,\mathrm{s}$ are not included in the datasets.
- 400 states
- 2 control inputs

A FMU [12] of the model or the generated dataset is available on request.

### A.5.8 MUJOCO DATASET

The MuJoCo dataset is provided through Minari (Younis et al., 2024) and is available as download[13]. The data is obtained from simulation of the *Gymnasium* (Towers et al., 2024) Ant Model[14]. The Ant model describes a 3D robot that consists of a torso with four legs where each leg consists of two parts, and the applied torques at the legs can be controlled.

---

[11]https://fmi-standard.org/
[12]https://fmi-standard.org/
[13]https://minari.farama.org/main/datasets/mujoco/ant/expert-v0/
[14]https://gymnasium.farama.org/environments/mujoco/ant/

The dataset consists of 2026 samples of different length, where the majority has a length of 1000. To make it compatible with our code, we select all sequences of length 1000 and have in total 1986 samples for training. The dataset has an observation space with 105 variables, which we assume to be the state space, and an action space of 8 variables, which is in our wording the control input space. From the state space, only the first 27 variables are proper system states, while the remaining 78 elements are external forces, which is why we exclude those from our training dataset.

### A.5.9 BENCHMARK NONLINEAR SURROGATE MODELS

The benchmark shows the RMSE normalized by respective variances of prediction of the surrogates for the dataset's states. For the Power Plant dataset, we predict the RMSE on output variables. The trained B-NODE models are constant variance B-NODEs with nonlinear layers. For the Latent ODE, we added to the ODE function of (Rubanova et al., 2019) an input vector, i.e.

$$\boldsymbol{x}^{z\,\prime}(t) = \mathbf{f}_{\boldsymbol{\phi}_{\mathrm{LNODE}}}(\boldsymbol{x}^z(t), \boldsymbol{u}^z(t)) \tag{40}$$

Dataset details for MuJoCo and Water Hammer can be found in A.5.8 and A.5.7.

### A.5.10 BENCHMARK LINEAR SURROGATE MODELS

The benchmark shows the RMSE normalized by respective variances of prediction of the surrogates for the dataset's states. As DMD methods are implemented for states prediction, we predict also states for the Power Plant dataset. The trained B-NODE models are constant variance B-NODEs with linear layers. Dataset details for MuJoCo and Small Grid can be found in A.5.8 and A.5.6. The DMD methods are implemented using the *PyKoopman*-package (Pan et al., 2024). For DMDc, we performed the algorithm for all ranks possible. For eDMDc, we employed radial basis functions as observables and implemented a genetic algorithm to search for best basis functions.

## A.6 ADDITIONAL EXEPERIMENT RESULTS

### A.6.1 MASKING OF IDENTIFIED LATENT CHANNELS

To evaluate if B-NODEs are able to learn a dimensionality reduced representation, that also holds when transforming the weakly defined information bottleneck (by KL divergence) into a hard one (by manually setting latent channels with $D_{\mathrm{KL},z_i} < 0.1$ to zero), we performed a training in which we set after sufficient training the mask. Specifically, we trained a nonlinear and linear B-NODE for SHF model. After setting the mask, $\beta 0$ is set to 0 and the B-NODE is trained as a NODE on a latent state space. As shown in Figure A.10, the error does not increase after masking the latent channels, it decreases additionally.

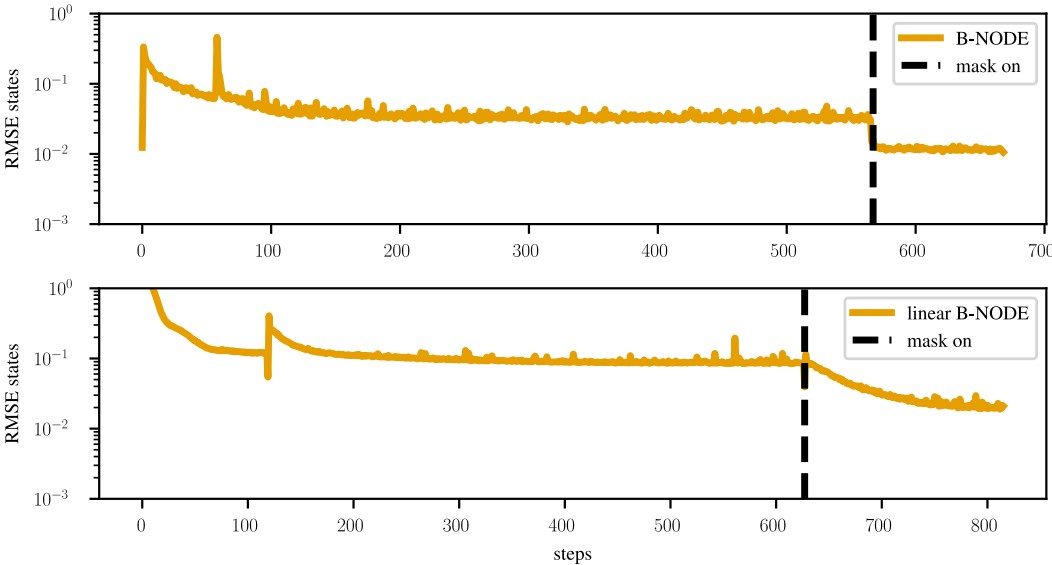

Figure A.10: Manually masking the identified latent channels to 0.

### A.6.2 HYPERPARAMETER SENSITIVITY OF STRATIFIED HEAT FLOW MODEL (SHF)

Table 4: Results of a hyperparameter sensitivity analysis for the stratified heat flow (SHF) model.

| Hyperparameter | Com. Test | Test | Train | States Test | Outputs Test | $n^z_{\boldsymbol{x},\text{used}}$ | $n^z_{\boldsymbol{u},\text{used}}$ | $n^z_{\boldsymbol{p},\text{used}}$ |
|---|---|---|---|---|---|---|---|---|
| **Baseline** | 0.0165 | 0.0165 | 0.0153 | 0.0010 | 0.0320 | **4** | **2** | **0** |
| n_linear_layers: 4 → 3 | 0.0171 | 0.0167 | 0.0166 | 0.0012 | 0.0323 | 4 | 2 | 0 |
| n_linear_layers: 4 → 5 | 0.0425 | 0.0431 | 0.0429 | 0.0021 | 0.0841 | 4 | 2 | 0 |
| linear_hidden_dim: 128 → 32 | 0.0180 | 0.0164 | 0.0169 | 0.0011 | 0.0317 | 4 | 2 | 0 |
| linear_hidden_dim: 128 → 64 | 0.0171 | 0.0162 | 0.0164 | 0.0012 | 0.0313 | 4 | 2 | 0 |
| **linear_hidden_dim: 128 → 256** | 0.0115 | 0.0110 | 0.0114 | 0.0011 | 0.0210 | **5** | **2** | **0** |
| activation: ELU → LeakyReLU | 0.0187 | 0.0178 | 0.0179 | 0.0025 | 0.0330 | 4 | 2 | 0 |
| activation: ELU → ReLU | 0.0462 | 0.0439 | 0.0432 | 0.0025 | 0.0853 | 3 | 2 | 0 |
| activation: ELU → Sigmoid | 0.9636 | 0.9459 | 0.9918 | 0.9531 | 0.9387 | 0 | 0 | 0 |
| activation: ELU → CELU | 0.0166 | 0.0158 | 0.0153 | 0.0010 | 0.0306 | 4 | 2 | 0 |
| activation: ELU → Tanh | 0.0174 | 0.0170 | 0.0168 | 0.0011 | 0.0328 | 4 | 2 | 0 |
| hidden_dim_output_nn: 128 → 32 | 0.0163 | 0.0150 | 0.0149 | 0.0009 | 0.0290 | 4 | 2 | 0 |
| hidden_dim_output_nn: 128 → 64 | 0.0160 | 0.0151 | 0.0158 | 0.0009 | 0.0294 | 4 | 2 | 0 |
| hidden_dim_output_nn: 128 → 256 | 0.0167 | 0.0147 | 0.0162 | 0.0009 | 0.0284 | 4 | 2 | 0 |
| controls_to_decoder: True → False | 0.3972 | 0.3803 | 0.3559 | 0.1475 | 0.6131 | 6 | 0 | 0 |
| lat_controls_dim: 12 → 6, lat_states_dim: 12 → 6 | 0.0162 | 0.0149 | 0.0155 | 0.0009 | 0.0289 | 4 | 2 | 0 |
| lat_controls_dim: 12 → 24, lat_states_dim: 12 → 24 | 0.0165 | 0.0156 | 0.0166 | 0.0010 | 0.0302 | 4 | 2 | 0 |
| lat_controls_dim: 12 → 64, lat_states_dim: 12 → 64 | 0.0165 | 0.0161 | 0.0161 | 0.0010 | 0.0311 | 4 | 2 | 0 |
| solver_atol_override: 0.0001 → 1e-05, solver_rtol_override: 0.001 → 0.0001 | 0.0163 | 0.0163 | 0.0179 | 0.0011 | 0.0315 | 4 | 2 | 0 |
| beta_start_override: 0.1 → 1.0 | 0.1407 | 0.1377 | 0.1470 | 0.0163 | 0.2591 | 3 | 2 | 0 |
| beta_start_override: 0.1 → 0.01 | 0.0035 | 0.0038 | 0.0036 | 0.0002 | 0.0074 | 5 | 2 | 0 |
| **beta_start_override: 0.1 → 0.001** | 0.0032 | 0.0031 | 0.0032 | 0.0002 | 0.0060 | **12** | **7** | **0** |

