# OpenReview forum: "Balanced Neural ODEs: nonlinear model order reduction and Koopman operator approximations"
_ICLR.cc/2025/Conference — ICLR 2025 Poster_

### Official Review · Reviewer_LoSy · 2024-10-29

**Soundness:** 3
**Presentation:** 2
**Contribution:** 2
**Rating:** 5
**Confidence:** 2

**Summary:**

This paper proposes a novel method called Balanced Neural ODE (B-NODE), which combines Variational Autoencoders (VAEs) with Neural ODEs to reduce the complexity of dynamical systems while maintaining accuracy. This approach extends Latent ODEs to handle time-varying inputs by continuously propagating latent variables. It also aims to approximate Koopman operators by leveraging the weight factor of KL divergence during VAE training. The proposed method is evaluated on both academic and real-world scenarios.

**Strengths:**

- The problem addressed is both interesting and crucial for applying Neural ODEs to dynamical systems with time-varying inputs. The combination of VAE and Neural ODEs to achieve dimensionality reduction alongside dynamic modeling provides an approach for managing complex systems.

- The inclusion of real-world test cases adds practical value to the study and demonstrates its potential applicability.

**Weaknesses:**

## Presentation
- The layout of this paper is somewhat disorganized, which makes it very hard to follow. The presentation is unclear without a main thread, with complex concepts that are not explained adequately and unnecessary.
- The concepts and some formulations appear to be justified solely by the authors themselves, lacking sufficient external validation or references to prior work (see points in Questions).
- Please ensure there is proper spacing between different paragraphs and sections.

## Experiments
- There are no baseline comparisons with Koopman-based models, Neural ODEs, or VAE-based forecasting approaches (which are highly relevant to this work), which limits the ability to assess how much improvement the proposed model actually provides.

## Minor issues
some examples:
- Lines 100 the integral should be $\int_{t_0}^{t}f_{\phi NODE}(\mathbf{x}(\tau))d\tau$
- Line 225, is -> are
- Line 312, Observed -> observe

I'd encourage authors redo proofreading.

**Questions:**

- In line 159, the statement "Combining VAEs with State Space Models leverages VAEs' ability to generalize through an information bottleneck with a numerically pre-determined latent space that promotes data locality and latent orthogonality" is unclear. Specifically, what is meant by "data locality" and "latent orthogonality"? Could the authors provide further explanation, relevant citations, or supportive numerical results to clarify these concepts?

- Could author further explain how they arrive equation (4ba)? You assume that the $x_0^z$ only related to initial state $x_0$?

-  Lines 297-299: Could the authors further justify why they do not need to pre-determine the dimensionality of the latent space? The weighting factor might contribute to sparsity in certain contexts, but it is unclear how it directly adapts the dimensionality. Additional explanation on this point would be helpful.

---

> ### Author Response · Authors · 2024-11-20
> **Answer part 1**
>
> Dear Reviewer,
>
> Thank you very much for the time you took and the comments you bring up that will help to improve our paper.
>
> ## Weaknesses
>
> ### Presentation
>
> (1) Concerning your first comment, could you name what you mean by “the  layout of the paper is somewhat disorganized”? Do you mean the structure (i.e. Introduction; Background, Method, Applications, Discussion, Conclusion), or do you mean how/where we inserted the figure? What caused your comment “The presentation is unclear without a main thread”? Do you think that two sentences in the Background section would help to understand why we explain NeuralODE and VAE there, apart from the first two sentences of the abstract?  Which complex concepts do you think are not explained adequately (dimensional reduction with VAE?), which concepts are unnecessary?
>
> (2) - - > answer in section questions
>
> (3) We used the ICLR template and did not adjust spacing between paragraphs and sections. Could you tell us at which position you see insufficient spacing?
>
> ### Experiments
>
> (1) First, I would like to reiterate the motivation for the ICLR conference “What is the significance of the work? Does it contribute new knowledge and sufficient value to the community? Note, this does not necessarily require state-of-the-art results. Submissions bring value to the ICLR community when they convincingly demonstrate new, relevant, impactful knowledge (incl., empirical, theoretical, for practitioners, etc).”  We believe that a big advantage of our work compared to other model-order methods (such as eDMDc) is the methodology itself; for learning (linear) surrogate models, we do not need to pre-determine bottleneck dimensions or choose appropriate basis functions. As it is a model-order reduction methods, its performance outcome is always a compromise between state reduction and prediction accuracy.
>
> I would like to explain why we cannot provide a comparison to the machine learning task of time series forecasting as in Koopman-based and VAE-based model you mentioned, e.g. in [1] and [2]. In our terminology, this would translate to predicting a time-series starting at an initial state $x^z(t_0)$, without control inputs $u(t), t>t_0$ and parameters $p$. In contrast to that, we aim for the generation of model-order reduced dynamical system models, that can predict the quantities of states $x(t)$ and output variables $y(t)$ of dynamical systems, given an initial state $x(t_0)$ and continuously arriving inputs signals $u(t), t>t_0$. This means, that forecasting of a time-series is not our target, our goal is building a dynamical system model. You might have noticed that we identify the initial latent state $x^z_0$ in equation 4ba just from the state vector $x(t_0)$. We obtain this state vector from the simulations of the physical model we aim to build a surrogate model for. We can do this for simulations; but not for real-world data. Control theory uses the term observability if one can determine the systems state. Observability does not mean that all states can be inferred from a snapshot of measurements at one time $t_0$. In general, important states might not be measured and can only be determined from a time-series $t_{-T_{obs.}:0}$ of system measurements $y(t)$. In control theory, state observers such as Luenberger observers and Kalman filters are therefore used. However, as we currently aim with B-NODEs for surrogate models that replace equation-based simulation models with neural-based models, we did not focus on developing a state observer. Additionally, we do also believe that the better scientific approach is to first develop a good method to learn dynamical models (potentially B-NODE). After that, we also figured that projecting the classical control theory concepts (Luenberger observer or Kalman filter) on the B-NODE context would be a certainly interesting area for future research. Then, a publication about that should for sure include a comparison to other baseline models for time-series forecasting e.g. as in [1,2]
>
> Unfortunately for now, no benchmarks are established for our targeted problem. Our method targets dynamical systems with inputs, whose models can be integrated by using an ODE solver and therefore can profit e.g. from adaptive-step size solvers to improve simulation speed and can be integrated industry relevant simulation framework such as Modelica, Dymola, Simulink etc.

---

> > ### Author Response · Authors · 2024-11-20
> > **Answer part 2**
> >
> > I would like to challenge your argument “There are no baseline comparisons with Koopman-based models, Neural ODEs, or VAE-based forecasting approaches”.  We compared our method not only to a State Space Neural ODE, but also to the commonly-used truncation method of Truncated Balanced Realization (TBR) in Figure 4(a). Additionally, we did also compare it to Latent ODE in Figure 3(b), and verified our Koopman approach with an analytical example in Figure 7. A limited amount of baseline methods and datasets is quiet common in literature about model order reduction, as in [4,5,6]. Applying such a method on a real-world simulation problem such as the Thermal Power Plant model with detailed 2-phase properties of water, 100+ states, unsteadiness and nonlinear loops is to our knowledge very rare in literature, most data-driven model order reduction literature spans around quiet simple toy models.
> >
> > However, we agree that the paper would be improved by integrating more experiments and will work in the next week on providing more evaluations. We plan to add at least two more experiments (e.g. from Modelica, or MuJoCo) and at least one more model order reduction method (eDMDc), as well as providing the results of Vanilla Latent ODE with inputs.
> >
> > Does this explanation help? Do you see anything arising from this, that would help to improve the paper’s quality?
> >
> > ### minor issues
> >
> > Thank you for raising attention to these mistakes. We’ll improve this in the updated version.
> >
> > ## Questions
> >
> > (1) We forgot to explain data locality. We cited the paper explaining this [7] in line 140. Data locality means that points that are close in data space are encoded close to each other in latent space as well, which helps the model to generalize and learn disentangled representations (i.e. that each variable describes one different aspect of the data). A more strict description of disentanglement is “orthogonality”, as described by [8]. We cited and mentioned this aspect in line 145-146. We will add an explanation on data locality. Do you have any more suggestions to avoid this confusion? We explained the aspects we mention in lines 159-182 in the background section, and therefore did not cite the relevant papers again. Do you think that providing a reference to the background section might help?
> >
> > (2) Yes, that’s right. We assume that the latent initial state is only related to the initial (physical) state. In a simulation model, the set of states $x(t)$ at time $t$ defines together with the system’s parameters and inputs $u(t)$ at time $t$ unambiguously the system’s states, i.e. one can unambiguously calculate from this all other quantities of the system. Explanations for this can be found in [9] or [10, p. 6]. As mentioned previously in my comment: “Control theory uses the term observability if one can determine the systems state. Observability does not mean that all states can be inferred from a snapshot of measurements at one time $t_0$. In general, important states might not be measured and can only be determined from a time-series $t_{-T_{obs.}:0}$ of system measurements $y(t)$. In control theory, state observers such as Luenberger observers and Kalman filters are therefore used”. If you want to use data of that you don’t know if the system’s state is measured, you would need to implement a state observer, as also mentioned in line 217.
> >
> > (3) We explained how dimensional reduction works for plain $\beta$-VAE in line 147-155. This also works for B-NODE, e.g. see Fig. 3(a). We might should stress more that higher $\beta$ values increases the dimensional reduction, while smaller values lead to the usage of more latent dimensions. Is this explanation helpful? Would a reference to lines 147-155 help the reader to understand it better?
> >
> > We thank you a lot for the time you took to read and understand our paper. We hope that we could clarify some points and are looking forward to your answer.

---

> > > ### Author Response · Authors · 2024-11-20
> > > **Literature**
> > >
> > > Literature
> > >
> > > [1] Naiman, I., Erichson, N. B., Ren, P., Mahoney, M. W., & Azencot, O. (2023). Generative modeling of regular and irregular time series data via koopman VAEs. arXiv preprint arXiv:2310.02619.
> > >
> > > [2] Wi, H., Shin, Y., & Park, N. (2024, March). Continuous-time Autoencoders for Regular and Irregular Time Series Imputation. In Proceedings of the 17th ACM International Conference on Web Search and Data Mining (pp. 826-835).
> > >
> > > [3] Laurent Girin, Simon Leglaive, Xiaoyu Bie, Julien Diard, Thomas Hueber, and Xavier AlamedaPineda. Dynamical variational autoencoders: A comprehensive review. CoRR, abs/2008.12595, 2020. URL https://arxiv.org/abs/2008.12595.
> > >
> > > [4] Bethany Lusch, J. Nathan Kutz, and Steven L. Brunton. Deep learning for universal linear embeddings of nonlinear dynamics. Nature Communications, 9(1):4950, November 2018. ISSN 2041-1723. doi:10.1038/s41467-018-07210-0.
> > >
> > > [5] Brunton, Steven L., Joshua L. Proctor, and J. Nathan Kutz. "Discovering governing equations from data by sparse identification of nonlinear dynamical systems." Proceedings of the national academy of sciences 113.15 (2016): 3932-3937.
> > >
> > > [6] Han, Yiqiang, Wenjian Hao, and Umesh Vaidya. "Deep learning of Koopman representation for control." 2020 59th IEEE Conference on Decision and Control (CDC). IEEE, 2020.
> > >
> > > [7] Christopher P. Burgess, Irina Higgins, Arka Pal, Loic Matthey, Nick Watters, Guillaume Desjardins, and Alexander Lerchner. Understanding disentangling in β-vae, 2018. URL https://arxiv. org/abs/1804.03599.
> > >
> > > [8] Michal Rolinek, Dominik Zietlow, and Georg Martius. Variational autoencoders pursue pca directions (by accident). In Proceedings of the IEEE/CVF Conference on Computer Vision and Pattern Recognition (CVPR), 6 2019. doi:10.1109/CVPR.2019.01269.
> > >
> > > [9] Francois E Cellier and Ernesto Kofman. Continuous system simulation. Springer Science & Business Media, 2006. doi:10.1007/0-387-30260-3.
> > >
> > > [10] Herbert Werner. Control System Theory and Design. 2022. https://collaborating.tuhh.de/ICS/ics-public/lecture-files/-/blob/master/CSTD/ControlSystemsTheoryAndDesign.pdf

---

> > > > ### Comment · Reviewer_LoSy · 2024-11-25
> > > >
> > > > Thank you to the authors for the detailed response and additional clarifications. I appreciate your efforts in addressing the feedback, which have helped me better understand the positioning of this paper. Please find my comments below:
> > > >
> > > > (1) **Presentation:** Thank you for clarifying the formatting issue. The lack of spacing between paragraphs contributed to the presentation concern. I highly recommend including proper spacing between paragraphs in the updated manuscript, though I leave this decision to the authors.
> > > >
> > > > (2) **Experimental Setup:** Your explanation has provided greater clarity, and I acknowledge the contributions made through empirical validation.
> > > >
> > > > (3) **Questions:** I appreciate your detailed responses, which have adequately addressed my concerns and questions. I encourage the authors to incorporate these clarifications in the updated manuscript, particularly those related to data locality and Equation (4ba).
> > > >
> > > > Overall, I have updated my score accordingly.

---

> > > > > ### Author Response · Authors · 2024-11-28
> > > > > **Updated version of paper available**
> > > > >
> > > > > Dear Reviewer,
> > > > >
> > > > > We just uploaded an updated version of our paper and would like to point your attention towards changes we made to adress points you outlined.
> > > > >
> > > > >
> > > > > - Data Locality: We added an explanation on data locality in line 147. Overall, we also updated the following lines concerning dimensional reduction with $\beta$-VAE, most imporant with more explicit definition of "active-ness" of states.
> > > > >
> > > > > - Equation 4ba (now 5ba): we added a sentence concerning your point in line 227.
> > > > >
> > > > > - Additionally, we added more experiments and comparison to other methods in table 1 and table 2, along with a more details on simulation speed improvement in line 395-400, and as well a comparision of methods in the discussion section starting in line 486.
> > > > >
> > > > > We thank you again for your feedback and hope that we can further improve your impression of our work with the improvements made. We would be happy if the additional changes we made result in an improvement of your score. We remain open for discussion or questions if any remain.
> > > > >
> > > > > Thank you!
> > > > >
> > > > > Best, the authors.

---

### Official Review · Reviewer_1CNC · 2024-11-03

**Soundness:** 3
**Presentation:** 2
**Contribution:** 3
**Rating:** 6
**Confidence:** 3

**Summary:**

This work presents a novel approach combining Variational Autoencoders (VAEs) and Neural ODEs to address model order reduction (MOR) for dynamic systems. The proposed model, named Balanced Neural ODE (B-NODE), utilizes VAEs for latent dimensionality reduction and Neural ODEs for capturing transient dynamics, resulting in a compact, efficient surrogate model that can dynamically adapt to the complexity of input data.

**Strengths:**

The paper effectively combines Variational Autoencoders (VAEs) and Neural ODEs, leveraging their complementary strengths for dimensionality reduction and transient dynamics modeling. This approach could significantly impact surrogate modeling in dynamic systems.

**Weaknesses:**

Although the paper acknowledges areas for future work, a more detailed discussion on scenarios where B-NODE might underperform would be helpful. Specifically, it would be beneficial to outline any cases where the model might struggle with particular types of inputs or dynamic systems.

**Questions:**

- Please provide more comprehansive understanding of β-value?  A discussion of the theoretical implications of different β values
- It seems that β-value is small does it mean that information bottleneck become negligible?
- if you set β=0 what will happen? please discuss the theoretical implications of setting β=0

If the authors can clarify my concern, I could change my opinion.

---

> ### Author Response · Authors · 2024-11-19
> **Answer**
>
> Dear Reviewer,
>
> Thank you a lot for your time and the valuable feedback.
>
> Regarding the weakness you outlined, can you provide maybe one or two examples for what you mean? From my point of view, our method is applicable to every system that exhibits dynamics that are possible to be represented in a continuous form, i.e. $\dot{x}=f(x,u)$.
>
> ## Questions
>
> As space in the paper is limited, we tried to keep the $\beta$-VAE explanation short and referred in line 140 to [1], and mentioned the $\beta$ values implications again in line 314 and 346.
>
> You are right in your assumptions. A large $\beta$ will increase the importance of transmitting the information with the least amount of information possible (i.e. lowest number of states), while a small $\beta$ value makes the pressure on a reduced-order representation less strong. This can be seen in Figure 5, where small beta values correspond to more states, but also less prediction error. If we set $\beta=0$, the B-NODE degrades to a NODE on a latent space without any control of the latent space’s structure. By this we mean that the KL-Divergence in the loss forces the latent state for $\beta$-values $>0$ numerically to be centered around 0. Actually, when plotting a large number of states from simulations of a trained B-NODE, we can see a normal distribution of states around 0. For $\beta=0$, the model will bring the variance values $\sigma$ to 0, and the degraded B-NODE will use all states without limitations, maybe leading to instabilities as the same system state might have different latent representations.
>
> We will try to add a section with more clear discussion of the implications of different values for the $\beta$ values.
>
> We hope that this could clarify your concern. If any questions remain, we are happy to answer them. If not, we would be happy if you could increase your rating.
>
> Thank you!
>
> [1] Christopher P. Burgess, Irina Higgins, Arka Pal, Loic Matthey, Nick Watters, Guillaume Desjardins, and Alexander Lerchner. Understanding disentangling in β-vae, 2018. URL https://arxiv. org/abs/1804.03599.

---

> > ### Comment · Reviewer_1CNC · 2024-11-25
> >
> > Thank you for your response, I've adjusted my score accordingly.

---

> > > ### Author Response · Authors · 2024-11-25
> > >
> > > Thank you very much for the time you took to review our paper and increasing our score! :)

---

> ### Author Response · Authors · 2024-11-28
> **Updated paper version available**
>
> Dear Reviewer,
>
> We just uploaded a revised version of our paper and would like to point your attention towards changes we made to adress weaknesses you outlined:
> - To better evaluate B-NODE's performance on different systems, we added a benchmarks in Table 1 for the nonlinear case an in Table 2 for the linear case. In addition to the stratified heat flow model, we added a dataset from the MuJoCo (the MuJoCo Ant),  a fluid system example (water hammer), and a CFD example (small grid). We also added comparision to other state of the art methods, apart from state space neural ODE and Latent ODE, for the linear case (eDMDc and DMDc).
>
> - We added more explicit explanation on the $\beta$-parameter, e.g. by adding the "direction" how $\beta$ scales the balance at multiple places in the text, e.g. line 147, 191. This influence can also be seen in Figure 4(a) and our added benchmarks in table 1 and 2. Motivated by your question what happens if we set $\beta=0$, we also added an experiment with $\beta=0$.
>
> We thank you again for your feedback and hope that we can further improve your impression of our work with the improvements made. We would be happy if the additional changes we made result in an improvement of your score.  We remain open for discussion or questions if any remain.
>
> Thank you!
> Best, the authors.

---

### Official Review · Reviewer_w89j · 2024-11-04

**Soundness:** 3
**Presentation:** 2
**Contribution:** 2
**Rating:** 6
**Confidence:** 3

**Summary:**

This paper proposes a novel method, called B-NODE, for time-series modeling of dynamical systems (e.g. power plants). B-NODE is claimed to be fast (in terms of runtime) and accurate (in terms of reconstruction error on the original data). The main idea behind B-NODE is a combination of $\beta$-VAE (for dimensionality reduction) with a Neural ODE (for dynamical systems' modeling).

The faster runtime of B-NODE results from dimensionality reduction via the VAE. Previous approaches (like standard latent ODEs) fail to provide such dimensionality reduction. B-NODE also claims to offer the user with a knob to balance the trade-off between dimensionality reduction and reconstruction accuracy.

As for the paper structure, section 3 describes the methodology. Sections 4 and 5 demonstrate the advantages of B-NODE on certain synthetic and real-world tasks.

**Strengths:**

- The paper shows promising results with using B-NODE on certain tasks.
- The paper provides certain handy illustrations and detailed figures explaining their approach.

**Weaknesses:**

**Technical Weaknesses**:
1. *Limited evaluation.* Since this is a paper that proposes a novel method to solve a problem, the authors should aim to provide a comprehensive suite of experiments showing the advantages of the proposed methodology over existing baselines. The only real-world use case is presented in Section 4.2, which is just one single task. In Figure 5 (in the same section), B-NODE is compared against only one baseline, which is NODE. As a reference, two closely related papers in this field are [1] and [2]. Both provide much more detailed evaluation (Table 3 in [1] and Tables 2,3 in [2]).

**Presentation Weaknesses**:

1. *Unclear on the main advantage of B-NODE.* Is your main contribution speed of the proposed methodology, or accuracy, or both? It would be good to explicitly state it. The paper abstract mentions the word "fast", but the only runtime comparison I could find was Figure 6(b), which is only discussed briefly in the text in lines 388-392. In contrast, Figures 4,5,7,8 all seem to focus on accuracy via the RMSE.

2. *Vague exposition.* Below I list certain examples where the exposition is vague.

Example 1: The "B" in B-NODE stands for "Balanced", but what exactly is the knob that lets one choose the balance of the trade-off between reconstruction accuracy and latent dimension size?

Example 2: In lines 150-155, how is the $D_{KL}$ measured per channel? The overall $D_{KL}[q(z|x^\mathcal{D})||p(z)]$ is well-defined, but to measure for certain channel $i$, do you measure $D_{KL}[q(z_i|x^\mathcal{D})||p(z_i)]$? Are the marginals on a "channel" $i$ well-defined? It would have been nice to see this in proper notation.

Example 3: It is unclear what the following statement means: "The VAEs sampling-based robustness complements stability enhancements for Neural ODEs". This statement is mentioned twice, in line 180 (Methodology section) and line 476 (Conclusion section).

Example 4: Line 280 says "state $x_i^z$ is inactive", but no notion of active/inactive-ness is defined? Likely by inactive, the authors mean that the $D_{KL}$ of the $i^{th}$ channel is below the threshold, but this is sloppy exposition.

Example 5: Line 60 (and a few other places) mention that B-NODE *requires* a non-hierarchical prior, but the paper does not explain why this is the case.

Overall, I found the paper difficult to understand. Sentences that follow one another do not necessarily build the same idea in a linear/sequential fashion. This makes it hard to extract what the authors are trying to convey.


```
[1] Naiman, I., Erichson, N. B., Ren, P., Mahoney, M. W., & Azencot, O. (2023). Generative modeling of regular and irregular time series data via koopman VAEs. arXiv preprint arXiv:2310.02619.
[2] Wi, H., Shin, Y., & Park, N. (2024, March). Continuous-time Autoencoders for Regular and Irregular Time Series Imputation. In Proceedings of the 17th ACM International Conference on Web Search and Data Mining (pp. 826-835).
```

**Questions:**

**Technical questions**:
1. How exactly does B-NODE "balance" between dimensionality reduction and reconstruction accuracy? Is it simply just varying the $\beta$ in $\beta$-VAE? If yes, it would be nice to explicitly mention this.
2. In equation 4, why is it $q(u_{0:T}^z|u_{0:T})$ instead of $q(u_{0:T}^z|u_{0:T}, p^z, p)$? Since the latter is what one would get from the definition of $x^\mathcal{D}$ and $z$ in line 188. Is it because the external input $u$ is independent of the time-invariant parameters $p$? Is this a standard assumption?

**Suggestions for presentation**:
1. In the first paragraph explaining motivation, it would be nice to present more examples so that the reader can conceptualize the problem. Currently, only the words "yearly energy system simulations" are mentioned as examples.
2. In Figure 4(a), the reader needs to zoom in a lot to read the values of $\beta$ around the markers on the plot. Please make the plots easier to read.

---

> ### Author Response · Authors · 2024-11-19
> **Answer part 1**
>
> Dear reviewer,
>
> Thank you, a lot, for your detailed feedback and the time you put into that. I believe that the points you outlined will help to increase our paper’s quality.
>
> First, considering all of the reviewers’ answers, I want to address a point that seems important to stress more and that we will outline better in updated version of the paper as well. In our work, we’re learning a model that can react to continuous-time input signals $u(t)$, in which it differs considerably from in machine learning commonly known models for time-series prediction. These methods infer some kind of state $x(t_0)$ at a time $t_0$, and then predict output signals $y(t)$ for $t>t_0$. In contrast, we learn surrogate models for physical simulation models, of that we can access the initial state $x_0$ from simulation data, such that there is no need to infer an initial state from a sequence of variables $x_{-T:0}, y_{-T:0},u_{-T:0}$. Then, by continuously incorporating the input signals $u(t)$ at $t>t_0$, we predict the systems quantities $x(t), y(t)$ at $t>t_0$.
>
> Concerning your suggestion for presentation (1), we will therefore update the paper with more examples to make the targeted use-case more graspable.
>
> In the following, I would first like to react to the weaknesses you outlined, and then answer your questions.
> ## answer to weaknesses
>
> ### technical weaknesses
> Concerning your point evaluation, I would first like to explain why we can not provide an evaluation with method and datasets as in your cited papers [1] and [2]. While writing our paper, we were aware of their work (as we did also cite it), but their methods and our methods aim for different things: [1] and [2] concern time series forecasting and filling gaps of irregularly sampled time-series. In our terminology, this would translate to predicting a time-series starting at an initial state $x^z(t_0)$, without control inputs $u(t), t>t_0$ and parameters $p$. In contrast to that, we aim for the generation of model-order reduced dynamical system models, that can predict the quantities of states $x(t)$ and output variables $y(t)$ of dynamical systems, given an initial state $x(t_0)$ and continuously arriving inputs signals $u(t), t>t_0$. This means, that forecasting of a time-series is not our target, our goal is building a dynamical system model. You might have noticed that we identify the initial latent state $x^z_0$ in equation 4ba just from the state vector $x(t_0)$. We obtain this state vector from the simulations of the physical model we aim to build a surrogate model for. We can do this for simulations; but not for real-world data. Control theory uses the term observability if one can determine the systems state. Observability does not mean that all states can be inferred from a snapshot of measurements at one time $t_0$. In general, important states might not be measured and can only be determined from a time-series $t_{-T_{obs.}:0}$ of system measurements $y(t)$. In control theory, state observers such as Luenberger observers and Kalman filters are therefore used. However, as we currently aim with B-NODEs for surrogate models for replacing equation-based simulation models with neural-based models, we did not develop a state observer. Additionally, we do also believe that the better scientific approach is to first develop a good method to learn dynamical models (potentially B-NODE). After that, we also figured that projecting the classical control theory concepts (Luenberger observer or Kalman filter) on the B-NODE context would be a certainly interesting area for future research. Then, a publication for time series forecasting is possible and should for sure include a comparison to other baseline models as in your mentioned publications [1,2].
>
> Unfortunately for now, no benchmarks are established for our targeted problem. [3] provides a review for neural-network based models for dynamical systems. Our method targets dynamical systems with inputs, whose models can be integrated by using an ODE solver and therefore can profit e.g. from adaptive-step size solvers to improve simulation speed and can be integrated industry relevant simulation framework such as Modelica, Dymola, Simulink etc. For this, only NeuralODEs and Neural Controlled Differential equations (NeuralCDE) are presented in the review [3]. However, both do not perform model-order reduction.

---

> > ### Author Response · Authors · 2024-11-19
> > **Answer part 2**
> >
> > I would like to challenge your argument “B-NODE is compared against only one baseline, which is NODE.”  We compared our method not only to a State Space Neural ODE, but also to the commonly-used truncation method of Truncated Balanced Realization (TBR) in Figure 4(a). Additionally, we did also compare it to LatentODE in Figure 3(b), and verified our Koopman approach with an analytical example in Figure 7. A limited amount of baseline methods and datasets is quiet common in literature about model order reduction, as in [4,10,11]. Applying such a method on a real world simulation problem such as the Thermal Power Plant model with detailed 2-phase properties of water, 100+ states, unsteadiness and nonlinear loops is to our knowledge very rare in literature, most literature spans around quiet simple toy models.
> > However, we will work in the next week on providing more evaluations. We plan to add at least two more experiments (e.g. from Modelica ,or MuJoCo) and at least one more model order reduction method (eDMDc), as well as providing the results of Vanilla Latent ODE with inputs.
> >
> > Would this remove all the technical weaknesses you mentioned? Is there something else could be done to overcome them?
> >
> > ### Presentation Weaknesses
> >
> > (1) Unclear main advantage: We thank you for the comment on this and are willing to improve this in the paper. As B-NODE is a model order reduction method, its prediction outcome is always a compromise between speed and accuracy. Typically, the state dimension scales the computational burden of a simulation problem, so reducing its dimensionality is crucial to increase speed of ODE simulations. To stress the importance of this point; implicit ODE solvers use the Jacobian of the state derivative vector, whose computational expense scales quadratically with the number of states. Because of this, the result in Fig. 5 is highly relevant – we can reduce the state dimension from ca. 100 to 6. An advantage of B-NODE in contrast to other deep learning based surrogate modeling methods [4,5,6] (see line 84) is that the bottleneck dimensionality - i.e. the state reduction - does not need to be pre-determined and is adequately learned during training, such that training with e.g. 2 $\beta$-values is sufficient to find a good compromise between speed and accuracy. Additionally, compared to NODE, speed is one advantage – B-NODE needs only about half the number of function evaluations for integrating the right-hand side of the ODE as the State Space Neural ODE for the power plant example (7,000 vs 13,000) (we will add this result to our paper), which is why we had a 2 times faster evaluation than NODE (see Fig 6(b)). This is due to “smoothing” the ODE function output with the adaptive noise on its input during training, such that variable step size solvers less frequently have to decrease the step size. This consequence of applying stochastic noise to NODE training is also explained in [7,8,9] (see line 272). Additionally, we experienced increased training stability for B-NODE. We did not report this as it is hard to evaluate but will try to provide supporting experiments in the updated version. We provided the RMSE values (which are normalized by each variable’s variance) as an intuitive metric to evaluate the accuracy of the method given these achievements and could also show that it outperforms Truncated Balanced Realization and State Space NeuralODE in terms of accuracy (TBR, see Fig. 4(a)).  To sum up, B-NODE’s main advantage is the efficient compromise between speed and accuracy.

---

> > > ### Author Response · Authors · 2024-11-19
> > > **Answer part 3**
> > >
> > > (2) Vague exposition:
> > >
> > > Example 1: It is the $\beta$ value. We tried to outline this in in line 143, 314 and 346, but will add this at other positions in the text. ( - - > see answer to question)
> > >
> > > Example 2: To deal with the space limit of the text, we moved the definition of $D_{KL}[q(z|x^\mathcal{D})||p(z)]$  to the appendix (equation 27) as we thought it was known by the community. For the assumption of a Gaussian with diagonal covariance, an analytical expression for $D_{KL}$  is found, which can be evaluated for every latent dimension individually (equation 27). We will make this more clear in the main text.
> > >
> > > Example 3: This is explained in line 272 with citations of [7,8,9], and the mechanism for incorporating of the noise is defined in equation (13), (14). What would you propose to make this better understandable? Linking this paragraph at the other positions in the text?
> > >
> > > Example 4: We provided a definition for measuring that a latent channel is active in line 155 and  showed this empirically in A3.3. Additionally, we tested if this definition holds by masking all latent channels to 0 that fulfilled this condition and did not see an increase in prediction quality. I believe that the confusion of 2 and 4 might be due to the fact that we introduced the VAE with $\boldsymbol{x}^\mathcal{D}$ and $\boldsymbol{z}$, and provided no indexes in line 147 following for $z$. We will get this right by providing indexes, i.e. $D_{KL}(z_i)$ etc. Would that help? Is the introduction of the latent and physical variables in line 188 clear?
> > >
> > > Example 5: We mentioned this in line 59 “To achieve measurable state reduction, our method requires a non-hierarchical prior. As a consequence, dynamics are part of the inference model, which additionally contains a temporal bottleneck, while the information bottleneck is represented by the latent space.” which might have been overseen by you. Do you think that repeating this statement at another position would help; or that this argument needs more explanation?
> > >
> > > We’re aiming for writing an understandable paper. Can you name examples where “Sentences that follow one another do not necessarily build the same idea in a linear/sequential fashion” is the case?
> > >
> > > ## Answers to Questions
> > >
> > > (1) Yes, you’re right. It is the $\beta$ parameter. We mentioned this in line 142 , 314 and 346, but will add it at more positions. It is hard to develop an understanding for $\beta$-VAE on the space we have in this paper, which is why we moved some parts of this to the appendix and referred to [12]. Do you think there’s another way to improve this?
> > >
> > > (2) This is actually a design decision of the framework. Your proposal to make the latent embeddings of control inputs depended on the system time-invariant parameters $\boldsymbol{p}$ is an interesting idea. We did also test this but found worse reconstruction quality.  These decisions are inductive biases as outlined in [14].
> > >
> > > ### Suggestions for presentation:
> > > (1) We will add more examples. Just to make this more graspable for now, we’d like to add examples: E.g. it is easy to make detailed heat pump simulation models that model the heat transfers in the different heat exchangers on detail, but are way to heavy to be used for yearly simulations, as these models are sometimes slower than real time. Another example are coupled system simulations in the automotive or aircraft industry, where each supplier provides components that model their component. For the performance and optimization of the overall system, detailed results of the single component is not of interest, which is why surrogate models are heavily requested industry right now.
> > >
> > > (2) We will increase font size.
> > >
> > > ### Thank you
> > > We thank you a lot for your valuable feedback regarding clarity of our publication, that will help to improve our paper’s quality. We believe that our method for generating model-order reduced surrogates models that does not rely on extensive hyperparameter tuning (such as other approaches) is of great value to the community. We hope that the updated version will help to clarify your concerns such that you can raise the rating.

---

> > > > ### Author Response · Authors · 2024-11-19
> > > > **Literature to answers**
> > > >
> > > > [1] Naiman, I., Erichson, N. B., Ren, P., Mahoney, M. W., & Azencot, O. (2023). Generative modeling of regular and irregular time series data via koopman VAEs. arXiv preprint arXiv:2310.02619.
> > > >
> > > > [2] Wi, H., Shin, Y., & Park, N. (2024, March). Continuous-time Autoencoders for Regular and Irregular Time Series Imputation. In Proceedings of the 17th ACM International Conference on Web Search and Data Mining (pp. 826-835).
> > > >
> > > > [3] Christian Møldrup Legaard, Thomas Schranz, Gerald Schweiger, Jan Drgona, Basak Falay, Claudio Gomes, Alexandros Iosifidis, Mahdi Abkar, and Peter Gorm Larsen. Constructing neural networkbased models for simulating dynamical systems, 2022. URL https://dl.acm.org/doi/10.1145/3567591.
> > > >
> > > > [4] Bethany Lusch, J. Nathan Kutz, and Steven L. Brunton. Deep learning for universal linear embeddings of nonlinear dynamics. Nature Communications, 9(1):4950, November 2018. ISSN 2041-1723. doi:10.1038/s41467-018-07210-0.
> > > >
> > > > [5] Emmanuel Menier, Sebastian Kaltenbach, Mouadh Yagoubi, Marc Schoenauer, and Petros Koumoutsakos. Interpretable learning of effective dynamics for multiscale systems, September 2023. URL https://arxiv.org/abs/2309.05812.
> > > >
> > > > [6] Andreas Mardt, Luca Pasquali, Hao Wu, and Frank Noe. VAMPnets for deep learning of molecular kinetics. Nature Communications, 9(1):5, January 2018. ISSN 2041-1723. doi:10.1038/s41467-017-02388-1.
> > > >
> > > > [7] Alexandra Volokhova, Viktor Oganesyan, and Dmitry Vetrov. Stochasticity in neural ODEs: An empirical study. In ICLR 2020 Workshop on Integration of Deep Neural Models and Differential Equations, 2019. URL https://openreview.net/forum?id=C4ydiXrYw.
> > > >
> > > > [8] Arnab Ghosh, Harkirat Singh Behl, Emilien Dupont, Philip H. S. Torr, and Vinay P. Namboodiri. STEER : Simple temporal regularization for neural odes. CoRR, abs/2006.10711, 2020. URL https://proceedings.neurips.cc/paper/2020/hash/a9e18cb5dd9d3ab420946fa19ebbbf52-Abstract.html.
> > > >
> > > > [9] Xuanqing Liu, Tesi Xiao, Si Si, Qin Cao, Sanjiv Kumar, and Cho-Jui Hsieh. How does noise help robustness? explanation and exploration under the neural sde framework. In Proceedings of the IEEE/CVF Conference on Computer Vision and Pattern Recognition (CVPR), 6 2020. doi:10.1109/CVPR42600.2020.00036.
> > > >
> > > > [10] Brunton, Steven L., Joshua L. Proctor, and J. Nathan Kutz. "Discovering governing equations from data by sparse identification of nonlinear dynamical systems." Proceedings of the national academy of sciences 113.15 (2016): 3932-3937.
> > > >
> > > > [11] Han, Yiqiang, Wenjian Hao, and Umesh Vaidya. "Deep learning of Koopman representation for control." 2020 59th IEEE Conference on Decision and Control (CDC). IEEE, 2020.
> > > >
> > > > [12] Christopher P. Burgess, Irina Higgins, Arka Pal, Loic Matthey, Nick Watters, Guillaume Desjardins, and Alexander Lerchner. Understanding disentangling in β-vae, 2018. URL https://arxiv. org/abs/1804.03599.
> > > >
> > > > [14] Laurent Girin, Simon Leglaive, Xiaoyu Bie, Julien Diard, Thomas Hueber, and Xavier AlamedaPineda. Dynamical variational autoencoders: A comprehensive review. CoRR, abs/2008.12595, 2020. URL https://arxiv.org/abs/2008.12595.

---

> > > > ### Comment · Reviewer_w89j · 2024-11-21
> > > > **Thank you for answering my questions**
> > > >
> > > > **My response below about the examples of vague exposition I raised:**
> > > >
> > > > Example 3: Thank you for pointing to line 272.
> > > > Example 5: The explanation is simply "To achieve measurable state reduction, our method requires a non-hierarchical prior." This does not justify it fully. You should also aim to explain why a hierarchical prior will NOT achieve measurable state reduction.
> > > >
> > > > **My response below about the 2 questions I raised:**
> > > >
> > > > (1) Yes would be great to highlight upfront that you control the tradeoff with $\beta$.
> > > > (2) If removing some conditional dependencies is a specific design choice, then that should also be properly explained in text in my opinion. The reader should not be expected to understand this design choice just from seeing eq (4).

---

> ### Author Response · Authors · 2024-11-28
> **Updated paper version uploaded**
>
> Dear Reviewer,
>
> We uploaded today our updated version and would like to draw your attention to changes we made, in that we adress weaknesses you outlined.
>
> - Concerning limited evaluation, we added more experiments and comparision to other state of the art methods for the nonlinear case in table 1 (line 424) and linear case in table 2 (476). For the nonlinear case, we also provided an example with a dataset from MuJoCo.
> - Concerning clarity on main advantage, we added a discussion on that and comparison to other methods in Section 6 (starting on line 486). We also added more details on the considerable increase of computation speed (32 times-102 times higher) in line 395-400, and information about the number of required solver ODE calls in Figure 6(b) below.
>
> Concerning the points about vague exposition, we made the following changes:
>
> - Explicit description of tuning knob $\beta$: We added the direction how $\beta$ scales the balance add multiple places in the text, e.g. line 147, 191. This influence can also be seen in Figure 4(a) and the added benchmarks in table 1 and 2.
> - Why non-hierachical prior?  Explanation added in line 159-160.
> - Defintion of state "active-ness" and definition of KL Divergence marginals per latent channels:  We moved an equation ((4)) from the appendix that defines $D_{KL,z_i}$ per latent channel and used this definition in the following text. To additionally show that this definition of active-ness holds, we added an experiments in the Appendix (p. 30), cited in line 354, that shows that we can transform the weakly defined information bottleneck into a "hard" one by masking all latent channels with $D_{KL,z_i}<0.1$. Overall, we made multiple changes improving lines 143-184, where we describe dimensional reduction with $\beta$-VAE
>
> Concerning your question:
> - We made our design choice for independence of input encoding from parameters explicit in line 211.
>
> Concerning your suggestions for presentation:
> - We added more examples to conceptualize the approached problem better in the motivation (line 33-39).
> - We improved font size in figures.
>
> We thank you again for your feedback that helped to significantly improve our work.
> We tried to adresss all the points you outlined and would be happy if you would adapt your assesement of our work.
> We remain open to questions and discussion if there are any points still open.
>
> Best, the authors

---

### Official Review · Reviewer_kk7L · 2024-11-04

**Soundness:** 4
**Presentation:** 2
**Contribution:** 3
**Rating:** 8
**Confidence:** 3

**Summary:**

A method for model order reduction using $\beta$-VAEs and state space NeuralODEs called **balanced neural ODE** is described and tested on a few examples. The hyperparameter $\beta$ controls the number of active latent space dimensions, i.e. the reduced dimension of balanced neural ODE. Two main methods of evolution are proposed for the latent VAE parameters: constant-variance and dynamic-variance. Koopman operator approximation is shown to be a special case of balanced neural ODE.

**Strengths:**

1. Method is fully described, intuitive to understand, many derivations are fully given
2. Balanced neural ODE outperforms latent neural ODE and is able to learn latent dynamics.
2. Numerical experiments demonstrate the breadth/diversity of application of the method.

**Weaknesses:**

1. Method combines two existing approaches (however, the combination is novel)
2. Presentation feels brisk at times (might be alleviated with an extra page)
3. The model involves many components with architecture choices, and there is not enough ablation on

**Questions:**

1. Are there other related works that combine model order reduction and latent space evolution, other than latent ODE?
2. How does the balanced neuralODE compare to Dynamic Mode Decomposition method (for linear operator learning)?

---

> ### Author Response · Authors · 2024-11-16
> **Answer to review**
>
> Dear Reviewer,
>
> Thank you very much for your work and your valuable feedback.
>
> Regarding the weaknesses you outlined, I agree that the presentation might feel brisk at some times, regarding the page limit of 9 pages. Neverltheless, we tried to fit in as much explanations of foundations as possible, instead of for example referring to the review in [1]. Regarding ablation on the architectural choices, I would like to add something here: With the information bottlenecks we force the model to learn a “minimal” and orthogonal latent representation of the system’s dynamics. By orthogonal we mean that VAEs can learn disentangled informations, in that each latent factor describes a different aspect of information [2,3]. We hypothesize that this avoids overfitting, as only the necessary information are transmitted, and additional information channels that might cause overfitting, cannot be used. However, we observed during training no overfitting, i.e. train, test and validation losses had always (approximately) the same value.  Additionally, we performed a hyperparameter sensitivity analysis that revealed that the method is robust against different architectural choices (number of layers, hidden dimension, activation function, prescribed latent dimensionality). Furthermore, we did choose the same hyperparameters for alle distinct networks, because hyperparameter tuning shouldn’t be a focus of this work. Would you think that including this kind of information might increase the paper’s quality (having in mind the page limit)?
>
> To your questions:
>
> After reading the reviewers answers, we feel like it’s important to stress again the fact that we’re learning a model that can react to continuous-time input signals $u(t)$, in which it differs considerably from in machine learning commonly known models for time-series prediction. These methods infer some kind of state $x(t_0)$ at a time $t_0$, and then predict output signals $y(t)$ for $t>t_0$. In contrast, we learn surrogate models for physical simulation models, of that we can access the initial state $x_0$ from simulation data, such that there is no need to infer an initial state from a sequence of variables $x_{-T:0}, y_{-T:0}, u_{-T:0}$. Then, by continuously incorporating the input signals $u(t)$ at $t>t_0$, we predict the systems quantities $x(t), y(t)$ at $t>t_0$.
>
> (1) Because of that, first, I want to clarify, the Latent ODE [4] was in the original publication not thought for incorporating continuous-time inputs. To my knowledge, there are no related works use latent variables (with an information bottleneck prescribed through KL-divergence) for model order reduction. There are works that use latent variable models (e.g. simple (not variational) Autoencoder based models), e.g. as described in the review of [5, p.62], but the latent space dimension is an important hyperparameter to be chosen.
>
> (2) I think this is a great question, as it stresses an advantage of our method. Classical DMD was initially developed for generated reduced order models of simulations with very fine spatial distribution (as it is the case in CFD). For generating surrogates of simulations of systems without fine discretization, one needs to extend the dimensionality of the observable space (i.e. the variables obtained by simulation) with additional observations, generated by basis functions, such that that fitting a linear operator can be performed upon these variables [5, p. 75]. Basis functions can be sines, cosines, linear combinations, polynomials, or radial basis functions. However, finding the right basis functions is an additional aspect of surrogate model generation, and for our approach, something like this is not needed.
> I hope we could answer your questions. If something should be hard to understand or new questions are raised, please don’t hesitate to write again.
>
> Thank you a lot!

---

> > ### Author Response · Authors · 2024-11-16
> > **Literature to answer to review**
> >
> > [1] Laurent Girin, Simon Leglaive, Xiaoyu Bie, Julien Diard, Thomas Hueber, and Xavier AlamedaPineda. Dynamical variational autoencoders: A comprehensive review. CoRR, abs/2008.12595, 2020. URL https://arxiv.org/abs/2008.12595.
> >
> > [2] Christopher P. Burgess, Irina Higgins, Arka Pal, Loic Matthey, Nick Watters, Guillaume Desjardins, and Alexander Lerchner. Understanding disentangling in β-vae, 2018. URL https://arxiv.org/abs/1804.03599.
> >
> > [3] Michal Rolinek, Dominik Zietlow, and Georg Martius. Variational autoencoders pursue pca directions (by accident). In Proceedings of the IEEE/CVF Conference on Computer Vision and Pattern Recognition (CVPR), 6 2019. doi:10.1109/CVPR.2019.01269.
> >
> > [4] Yulia Rubanova, Ricky T. Q. Chen, and David K Duvenaud. Latent ordinary differential equations for irregularly-sampled time series. In H. Wallach, H. Larochelle, A. Beygelzimer, F. d'Alche-Buc, E. Fox, and R. Garnett (eds.), ´ Advances in Neural Information Processing Systems, volume 32. Curran Associates, Inc., 2019. URL https://proceedings.neurips.cc/paper_files/paper/2019/file/42a6845a557bef704ad8ac9cb4461d43-Paper.pdf.
> >
> > [5] Steven L. Brunton, Marko Budisic, Eurika Kaiser, and J. Nathan Kutz. Modern koopman theory for dynamical systems, 2021. URL https://arxiv.org/abs/2102.12086.

---

> ### Author Response · Authors · 2024-11-28
> **Updated Paper Version available**
>
> Dear Reviewer,
>
> We uploaded today our updated version and would like to draw your attention to changes we made, that also adressed weaknesses you outlined. Concerning weakness (2), we tried to improve the presentation clarity by reformulating and adding several sentences. Most significantly, we added 143-184 more detailed explanation of dimensional reduction with $\beta$-VAE.  We also added a more detailed motivation starting in line 33. Concerning weakness (3), we added on the last page of the paper a small hyperparameter sensitvity study.
> As pointed out in your questions, we also compared our method to DMD and and addded a comparision to other methods in the discussion section starting in line 486.
>
> We thank you again for your feedback and hope that we can further improve your impression of our work with the improvements made.
>
> Best, the authors.

---

### Meta-Review · Area_Chair_9EhN · 2024-12-21

**Metareview:**

The authors introduce a new method called Balanced Neural ODEs (B-NODE) which combine $\beta$-VAE and neural ODEs for modelling dynamical systems. Two primary applications are considered: state reduction and Koopman operator approximation. Reviewers appreciated the impressive benchmarks, showing B-NODE outperforming latent neural ODEs. While the presentation of the work was found to be quite dense, leading to some vague exposition, reviewers and authors were in agreement that the increased page limit and provided feedback should improve this aspect. This is good work, and should be in a state suitable for publication, provided that reviewer feedback is incorporated to improve the presentation.

**Additional Comments On Reviewer Discussion:**

Prior to the discussion phase, reviewers had concerns regarding the presentation of the work, and comparisons to suitable baselines, including models involving Koopman operators. Reviewer kk7L provided a highly positive review. Reviewer w89j was unclear on the main advantage of B-NODE, disliked some of the vague exposition, and preferred some additional evaluation. These issues were addressed in a revision, and the reviewer subsequently improved their score. Reviewer 1CNC asked about the effect of the beta value, and areas where B-NODE might fail. After the author response, the reviewer also improved their score. Reviewer LoSy gave a low score initially citing presentation concerns and issues with the experimental setup. The reviewer improved their score slightly during the discussion period, acknowledging some of their issues had been addressed.

---

### Decision · Program_Chairs · 2025-01-22

Accept (Poster)